# Machine learning of cloud types in satellite observations and climate models

Peter Kuma[1], Frida A.-M. Bender[1], Alex Schuddeboom[2], Adrian J. McDonald[2], and Øyvind Seland[3]

[1]Department of Meteorology (MISU), Stockholm University, Stockholm, Sweden
[2]School of Physical and Chemical Sciences, University of Canterbury, Christchurch, Aotearoa New Zealand
[3]Norwegian Meteorological Institute, Oslo, Norway

**Correspondence:** Peter Kuma (peter.kuma@misu.su.se)

**Abstract.** Uncertainty in cloud feedbacks in climate models is a major limitation in projections of future climate. Therefore, evaluation and improvement of cloud simulation is essential to ensure the accuracy of climate models. We analyse cloud biases and cloud change with respect to global mean near-surface temperature (GMST) in climate models relative to satellite observations, and relate them to equilibrium climate sensitivity, transient climate response and cloud feedback. For this purpose, we develop a supervised deep convolutional artificial neural network for determination of cloud types from low-resolution (2.5°×2.5°) daily mean top of atmosphere shortwave and longwave radiation fields, corresponding to the World Meteorological Organization (WMO) cloud genera recorded by human observers in the Global Telecommunication System (GTS). We train this network on top of atmosphere radiation retrieved by the Clouds and the Earth's Radiant Energy System (CERES) and GTS, and apply it on the Climate Model Intercomparison Project phase 5 and 6 (CMIP5 and CMIP6) model output and the ECMWF Reanalysis version 5 (ERA5) and the Modern-Era Retrospective Analysis for Research and Applications version 2 (MERRA-2) reanalyses. We compare the cloud types between models and satellite observations. We link biases to climate sensitivity and identify a negative linear relationship between the root mean square error of cloud type occurrence derived from the neural network and model equilibrium climate sensitivity (ECS), transient climate response (TCR) and cloud feedback. This statistical relationship in the model ensemble favours models with higher ECS, TCR and cloud feedback. However, this relationship could be due to the relatively small size of the ensemble used or decoupling between present-day biases and future projected cloud change. Using the abrupt-4xCO2 CMIP5 and CMIP6 experiment, we show that models simulating decreasing stratiform and increasing cumuliform clouds tend to have higher ECS than models simulating increasing stratiform and decreasing cumuliform clouds, and this could also partially explain the association between the model cloud type occurrence error and model ECS.

## 1 Introduction

Clouds are a major factor influencing the Earth's climate. They are highly spatially and temporally variable, with the top of atmosphere (TOA) radiation being particularly sensitive to cloud changes due to their high albedo and impact on longwave radiation. Of all climate feedbacks, cloud feedback is the most uncertain feedback in Earth system models (ESMs) (Zelinka et al., 2020; Sherwood et al., 2020). Therefore, it is essential that climate models converge more on a correct representation

of future clouds, but also on their representation of present-day clouds, which is a necessary (but not sufficient) condition for the fidelity of projected cloud change. The estimate of the 'likely' range (66%) of ECS has recently been refined in the 6<sup>th</sup> Assessment Report (AR6) of the Intergovernmental Panel on Climate Change (IPCC) to 2.5–4 K, from 1.5–4.5 K in AR5. Evidence for this estimate is only indirectly informed by the Climate Model Intercomparison Project phase 6 (CMIP6) models (Eyring et al., 2016, 2019), which have a multi-model mean of 3.7 K (Meehl et al., 2020). The combined assessment is based on paleoclimate and historical evidence, emergent constraints and process understanding. Notably, CMIP6 models predict a 16% higher multi-model mean than CMIP5 (3.2 K) (Meehl et al., 2020; Forster et al., 2020), and this fact has already been examined in a number of studies (Zelinka et al., 2020; Wyser et al., 2020; Schlund et al., 2020; Dong et al., 2020; Nijsse et al., 2020; Flynn and Mauritsen, 2020). It is also higher than the combined assessment central value of ECS of 3 K in AR6. The multi-model spread in CMIP6 is also larger than in CMIP5, with a standard deviation of 1.1 K in CMIP6 vs. 0.7 K in CMIP5. Modelling groups have prevailingly reported that the higher multi-model mean is due to changes in cloud representation in the recent generation of models (Meehl et al., 2020, Table 3), supported by the findings of Zelinka et al. (2020).

Recent understanding of climate sensitivity is represented by diverging results relative to the CMIP6 multi-model mean. Some authors have concluded that the high-ECS CMIP6 models are on average overestimating the ECS (Nijsse et al., 2020) and are not compatible with paleoclimatic records (Zhu et al., 2020, 2021). The high-ECS models are also not supported by the review studies of Sherwood et al. (2020) and AR6. In contrast, Bjordal et al. (2020) argue that high ECS models might be plausible because of state-dependent cloud phase feedback in the Southern Ocean. Models which simulate too much ice in the Southern Ocean clouds, a common bias among CMIP models, are expected to have lower cloud feedbacks globally because of a spuriously enhanced negative feedback associated with cloud phase changes in that region. Recently, Volodin (2021) reported that changing cloud parametrisation in the Institute of Numerical Mathematics Coupled Model version 4.8 (INM-CM4-8) from Smagorinsky type to a prognostic type of Tiedtke (1993) resulted in more than doubling of ECS from 1.8 K to 3.8 K. This underscores the importance of cloud parametrisation in determining model climate sensitivity. Jiménez-de-la-Cuesta and Mauritsen (2019) and Tokarska et al. (2020) estimate low ECS based on the historical record, Renoult et al. (2020) estimate low ECS based on paleoclimatic evidence from the last glacial maximum and mid-Pliocene warm period. Zhao et al. (2016) showed that it is possible to modify parametrisation of precipitation in convective plumes in the GFDL model and get different Cess climate sensitivities without increasing CRE error relative to CERES. Zhu et al. (2022) showed that in CESM2, a CMIP6 model with very high ECS of 6.1 K, a physically more consistent cloud microphysics parametrisation reduced the ECS to about 4 K and produced results more consistent with the last glacial maximum.

The effect of clouds on the climate comes primarily from cloud fraction and cloud optical depth, which are determined by factors and properties such as convection, mass flux, turbulence, atmospheric dynamics, cloud microphysics (cloud phase, cloud droplet and ice crystal size distribution, number concentration, ice crystal habit), vertical cloud overlap, cloud altitude, cloud cell structure, cloud lifetime and more. Accurate simulation of cloud within climate models is difficult not only because of the large number of properties, many of which are subgrid-scale in today's general circulation models, but also because compensating model biases may produce correct cloud radiative effect (CRE) while simulation of the individual properties

is incorrect. This may be especially true for global radiation budget, as model processes are often tuned to achieve a desired radiation balance at TOA (Hourdin et al., 2017; Schmidt et al., 2017).

Cloud genera (WMO, 2021a) have been an established way of describing clouds for over a century. They broadly correspond or correlate with the individual cloud properties such as cloud altitude, optical depth, phase, overlap and cell structure. Therefore, they can be used as a metric for model evaluation which, unlike metrics based on more synthetically-derived cloud classes, is easy to understand and has a very long observational record. So far, however, it has not been possible to identify cloud genera in low-resolution model output, because their identification depends on a high-resolution visual observation, generally from the ground. Here, we show that it is possible to use a supervised deep convolutional artificial neural network (ANN) to identify cloud genera in low-resolution model output and satellite observations. Past classifications of cloud types or cloud regimes derived from satellite datasets have been based on cloud optical depth and cloud top pressure or height by simple partitioning (Rossow and Schiffer, 1991) or by statistical clustering algorithms (Jakob and Tselioudis, 2003; McDonald et al., 2016; Oreopoulos et al., 2016; Cho et al., 2021), and on active radar and lidar sensors (Cesana et al., 2019), which likely only broadly correspond to human-observed cloud genera. More recently, deep ANNs have begun to be used to identify and classify clouds (Zantedeschi et al., 2020). Olsson et al. (2004) developed an ANN for classifying clouds in a numerical weather forecasting model output based on reference satellite data from the Advanced Very High Resolution Radiometer (AVHRR).

We introduce a new method of quantifying cloud types corresponding to the WMO cloud genera in model and satellite data based on an ANN approach. Furthermore, we quantify their global distribution and change with respect to global mean near-surface temperature (GMST) in CMIP5 and CMIP6 models, the Clouds and the Earth's Radiant Energy System (CERES) satellite data and two reanalyses, European Centre for Medium-Range Weather Forecasts's (ECMWF) reanalysis ERA5 (Hersbach et al., 2020) and the Modern-Era Retrospective Analysis for Research and Applications Version 2 (MERRA-2) (Gelaro et al., 2017). Convolutional artificial neural networks have been used before for cloud detection: Shi et al. (2017), Ye et al. (2017), Wohlfarth et al. (2018), Zhang et al. (2018), Liu and Li (2018) and Zantedeschi et al. (2020) used a convolutional ANN for identification of cloud genera in ground-based cloud images, and Drönner et al. (2018), Shendryk et al. (2019), Guo et al. (2020), Segal-Rozenhaimer et al. (2020) and Liu et al. (2021) developed a convolutional ANN for detecting cloudy pixels in high-resolution satellite imagery. While the determination of cloud types in model and satellite data and application of ANNs to identify cloud types are not new, in contrast to previous methods, we utilise cloud types with a direct correspondence to the established human-observed World Meteorological Organization (WMO) cloud genera to train our ANN. This dataset contains many decades of global cloud observations, recorded several times daily at a large number of stations. For this purpose, we develop an ANN which can be applied to input with low spatial and temporal resolution ($2.5°$, daily mean). This is because most current climate models provide output with low resolution. This resolution is not sufficient to represent individual clouds, but these can still be inferred statistically from large scale patterns. Likewise, the resolution of some satellite datasets such as CERES is on this spatial scale. We try to answer the question of whether cloud type biases and change with respect to GMST are related to cloud feedback, ECS and TCR in the CMIP models. The ANN and the associated code is made available under an open source license (Kuma et al., 2022).

## 2  Data

### 2.1  Satellite observations

We used satellite observations from the Clouds and the Earth's Radiant Energy System (CERES) in years 2003–2020 (Wielicki et al., 1996; Doelling et al., 2013; Loeb et al., 2018) as a reference training dataset for the ANN, and in particular, the daily mean adjusted all-sky and clear sky shortwave and longwave fluxes at TOA and shortwave (solar) insolation from SYN1deg 1°-resolution geostationary-enhanced product Terra+Aqua Edition 4.1. For evaluation of cloud top pressure and cloud optical depth, we used satellite-retrieved cloud visible optical depth (from 3.7 μm particle size retrieval) and cloud top pressure from the same product.

### 2.2  Climate models

CMIP5 and CMIP6 are the last two iterations of standardised global climate model experiments (Taylor et al., 2012; Eyring et al., 2016). We applied our ANN to the publicly-available model output of the *historical* and *abrupt-4xCO2* CMIP experiments in the daily mean products. An exception were the EC-Earth and NorESM2-LM models, which did not provide the necessary variables. For EC-Earth, we used data from the *hist-1950* experiment (model EC-Earth3P) of the High Resolution Model Intercomparison Project (HighResMIP) (Haarsma et al., 2016, 2020) as a substitute for historical. The model output resolution of EC-Earth3P is the same as EC-Earth. For NorESM2-LM, we obtained the data directly from the model developers. The variables used in our analysis were (exclusively) *rsut* (TOA outgoing shortwave radiation), *rlut* (TOA outgoing longwave radiation), *rsutcs* (TOA outgoing clear-sky shortwave radiation), *rlutcs* (TOA outgoing clear-sky longwave radiation), *rsdt* (TOA incident shortwave radiation) and *tas* (near-surface air temperature). In connection to CMIP models, we used estimates of the model ECS, TCR and cloud feedback from AR6, with missing values supplemented by Meehl et al. (2020), Zelinka et al. (2020), and ECS and TCR calculated with the ESMValTool version 2.4.0 (Righi et al., 2020). Here, we use a definition of cloud feedback adjusted for non-cloud influences as in Zelinka et al. (2020), Soden et al. (2008) and Shell et al. (2008). Table 1 lists CMIP5 and CMIP6 models used in our analysis, their ECS, TCR and cloud feedback. In total, we analysed 4 CMIP5 and 20 CMIP6 models, of which 18 had the necessary data in the historical experiment for comparison with CERES (years 2003–2014), and 22 had the necessary data in the abrupt-4xCO2 experiment. No selection was done on the models, i.e. all CMIP5 and CMIP6 models which provided the required fields in the CMIP archives were analysed here. For some models, ECS, TCR or cloud feedback were not available. For these models, the values were taken from a related available model (as detailed in Table 1). The model developers of IPSL-CM6A-LR-INCA advised us that its TCR should be the same as IPSL-CM6A-LR (Olivier Boucher, personal communication).

### 2.3  Reanalyses

In addition to CMIP, we analysed the output of two reanalyses: ERA5 (Hersbach et al., 2020) and MERRA-2 (Gelaro et al., 2017). From MERRA-2, we used the *M2T1NXRAD* product: daily means of the variables *LWTUP* (upwelling longwave flux at

**Table 1.** Table of CMIP5, CMIP6 models and reanalyses used in our analysis and their CMIP phase, equilibrium climate sensitivity (ECS), transient climate response (TCR), cloud feedback (CLD), if they provided the necessary variables in the *historical* (hist.) (in the case of reanalyses *historical reanalysis*) and *abrupt-4xCO2* experiments ('●' – yes, '○' – no, '-' – not applicable), and model output resolution (Res.) as the number of longitude × latitude bins. Models are sorted by their ECS. ECS, TCR and CLD were sourced from AR6, Zelinka (2021) and Semmler et al. (2021).

| # | Model | Phase | ECS (K) | TCR (K) | CLD (Wm$^{-2}$K$^{-1}$) | hist. | abrupt-4xCO2 | Res. (lon.×lat.) |
|---|---|---|---|---|---|---|---|---|
| 1 | INM-CM4-8 | 6 | 1.83 | 1.33 | -0.09 | ● | ● | 180×120 |
| 2 | INM-CM5-0 | 6 | 1.92 | 1.3 | -0.06 | ● | ● | 180×120 |
| 3 | NorESM2-LM | 6 | 2.54 | 1.48 | 0.44 | ● | ● | 144×96 |
| 4 | MRI-CGCM3 | 5 | 2.60 | 1.60 | 0.28 | - | ● | 320×160 |
| 5 | MPI-ESM-1-2-HAM | 6 | 2.96 | 1.8 | -0.16 | ● | ● | 192×96 |
| 6 | MPI-ESM1-2-HR | 6 | 2.98 | 1.66 | 0.27 | ● | ● | 384×192 |
| 7 | MPI-ESM1-2-LR | 6 | 3.00 | 1.84 | 0.18 | ● | ● | 192×96 |
| 8 | MRI-ESM2-0 | 6 | 3.15 | 1.64 | 0.46 | ● | ● | 320×160 |
| 9 | AWI-ESM-1-1-LR | 6 | 3.29 | 2.11 | 0.29[a] | ● | ○ | 192×96 |
| 10 | MPI-ESM-LR | 5 | 3.63 | 2.00 | 0.44 | - | ● | 192×96 |
| 11 | IPSL-CM5A2-INCA | 6 | 3.79 | 1.9 | 1.05 | ● | ● | 96×96 |
| 12 | GFDL-CM4 | 6 | 3.89 | 2.10 | 0.64 | ○ | ● | 144×90 |
| 13 | IPSL-CM5A-MR | 5 | 4.12 | 2.00 | 1.25 | - | ● | 144×143 |
| 14 | IPSL-CM5A-LR | 5 | 4.13 | 2.00 | 1.18 | - | ● | 96×96 |
| 15 | IPSL-CM6A-LR-INCA | 6 | 4.13 | 2.32[a] | 0.43 | ● | ● | 144×143 |
| 16 | CNRM-CM6-1-HR | 6 | 4.28 | 2.48 | 0.59 | ● | ● | 720×360 |
| 17 | EC-Earth3P | 6 | 4.31[a] | 2.62[a] | 0.37[a] | ● | ○ | 512×256 |
| 18 | IPSL-CM6A-LR | 6 | 4.56 | 2.32 | 0.45 | ● | ● | 144×143 |
| 19 | CNRM-ESM2-1 | 6 | 4.76 | 1.86 | 0.63 | ○ | ● | 256×128 |
| 20 | CNRM-CM6-1 | 6 | 4.83 | 2.14 | 0.61 | ● | ● | 256×128 |
| 21 | UKESM1-0-LL | 6 | 5.34 | 2.79 | 0.87 | ● | ● | 192×144 |
| 22 | HadGEM3-GC31-MM | 6 | 5.42 | 2.58 | 0.91 | ● | ● | 432×324 |
| 23 | HadGEM3-GC31-LL | 6 | 5.55 | 2.55 | 0.84 | ● | ● | 192×144 |
| 24 | CanESM5 | 6 | 5.62 | 2.74 | 0.88 | ● | ● | 128×64 |
| 25 | ERA5 | - | - | - | - | ● | - | 1440×721 |
| 26 | MERRA-5 | - | - | - | - | ● | - | 576×361 |

[a]For some models, ECS, TCR or CLD were not available. For these models, the values were taken from a related available model (CLD of AWI-ESM-1-1-LR as in AWI-CM-1-1-MR; ECS, TCR and CLD of EC-Earth3P as in EC-Earth3-Veg; and TCR of IPSL-CM6A-LR-INCA as in IPSL-CM6A-LR).

TOA), *LWTUPCLR* (upwelling longwave flux at TOA assuming clear sky), *SWTDN* (TOA incoming shortwave flux), *SWTNT*
(TOA net downward shortwave flux) and *SWTNTCLR* (TOA net downward shortwave flux assuming clear sky). From ERA5,
we used the ERA5 hourly data on single levels from 1979 to present product variables *tsr* (top net solar radiation), *tsrc* (top net
solar radiation, clear sky), *ttr* (top net thermal radiation) and *ttrc* (top net thermal radiation, clear sky). The variables were used
in an equivalent way to the CMIP5 and CMIP6 variables (Section 2.2).

## 2.4 Station observations

In addition to satellite and model data, we used ground-based land and marine station data from the Historical Unidata Internet
Data Distribution (IDD) Global Observational Data (Unidata, 2003). This dataset is a collection of the Global Telecommu-
nication System (GTS) (WMO, 2021b) reports, which come from synoptic messages sent by stations to the WMO network.
They consist of standard synoptic observations. For stations with an observer, clouds are identified visually at three different
levels (low, middle and high). For each level, a cloud genus/species category is recorded as a number between 0 and 9 or as not
available. Therefore, for each station, up to three numbers are available encoding cloud genera/species, the meaning of which is
explained in the WMO Manual on Codes (WMO, 2011) in the code tables 0509, 0513 and 0515. Only one cloud genus/species
category can be recorded for each level. Cloud fraction information in the station data was not used in our analysis.

The IDD records were available between 2003-05-19 and 2020-12-31 at standard synoptic times (00Z, 03Z, . . . , 21Z). We
excluded years in which more than three weeks of data were missing: 2006, 2007 and 2008. We used the cloud genus variables
of the synoptic (SYNOP) and marine (BUOY) reports: low cloud (IDD variable 'cloudLow') based on the WMO *Code Table
0513*, middle cloud based on *Code Table 0515* (IDD variable 'cloudMiddle') and high cloud based on *Code Table 0509* (IDD
variable 'cloudHigh') (WMO, 2011). Furthermore, we grouped the cloud genera/species into four cloud types to simplify our
analysis:

1. *high*: cirrus, cirrostratus, cirrocumulus ($C_H$ codes 1–9),

2. *middle*: altostratus, altocumulus ($C_M$ codes 1–9),

3. *cumuliform*: cumulus, cumulonimbus ($C_L$ codes 1–3, 8, 9),

4. *stratiform*: stratocumulus, stratus ($C_L$ codes 4–7).

To provide more detail, we also used an extended grouping of 10 cloud types:

1. *Ci*: cirrus ($C_H$ codes 1–6)

2. *Cs*: cirrostratus ($C_H$ codes 7–8)

3. *Cc*: cirrocumulus ($C_H$ code 9)

4. *As*: altostratus ($C_M$ codes 1–2)

5. *Ac*: altocumulus ($C_M$ codes 3–9)

6. *Cu*: cumulus ($C_L$ codes 1–3)

7. *Sc*: stratocumulus ($C_L$ codes 4–5)

8. *St*: stratus ($C_L$ codes 6–7)

9. *Cu+Sc*: cumulus and stratocumulus ($C_L$ code 8)

10. *Cb*: cumulonimbus ($C_L$ code 9)

As an example of the geographical distribution of stations, Fig. 1a shows the location of SYNOP and BUOY station reports with cloud data available on 1 January 2010 (Fig. 1b, c are discussed in Sect. 3.2). Because the data come from operational weather stations, they are geographically biased to certain locations, especially land, extratropics and the Northern Hemisphere. Undersampled locations are ocean, the Southern Hemisphere and the polar regions. Cloud type information from stations in the USA and Australia is also not available in the WMO records. Because of partially missing data in 2003, 2006 and 2008, we excluded these years in the ANN training phase. Not all stations in the IDD database provide cloud type information, and such stations were excluded from our analysis. High and middle clouds can be obscured by underlying cloud layers. In such cases, the observation of high or middle clouds is recorded as missing in the IDD data and we exclude such stations from the calculation of statistics for the middle or high cloud types, respectively. This limitation of the dataset means it is less suitable for identifying middle and high clouds than low clouds in multi-layer cloud situations, and a similar, but reverse, limitation exists in spaceborne cloud observations (McErlich et al., 2021).

## 2.5 Historical global mean near-surface temperature

Historical GMST was sourced from the NASA Goddard Institute for Space Studies (GISS) Surface Temperature Analysis version 4 (GISTEMP v4) (Lenssen et al., 2019; GISTEMP Team, 2021). This dataset was used in combination with the CERES dataset to determine observed change of cloud type occurrence with respect to GMST.

## 3 Methods

### 3.1 Rationale and methods outline

We trained an ANN on satellite observations from the Clouds and the Earth's Radiant Energy System (CERES) and ground based observations from WMO stations in the Internet Data Distribution (IDD) dataset (Unidata, 2003). Then, we applied the ANN on CERES data, CMIP5 and CMIP6 model output, ERA5 and MERRA-2. A large database of ground-based cloud observations has been compiled in the IDD dataset. This database contains human-observed cloud information at standard synoptic times, encoded as three numbers between 0 and 9 (or missing) for low, mid-level and high clouds specifying the cloud genus and species (WMO, 2011, 2021a). ANNs are typically used for various forms of image labelling, where the ANN

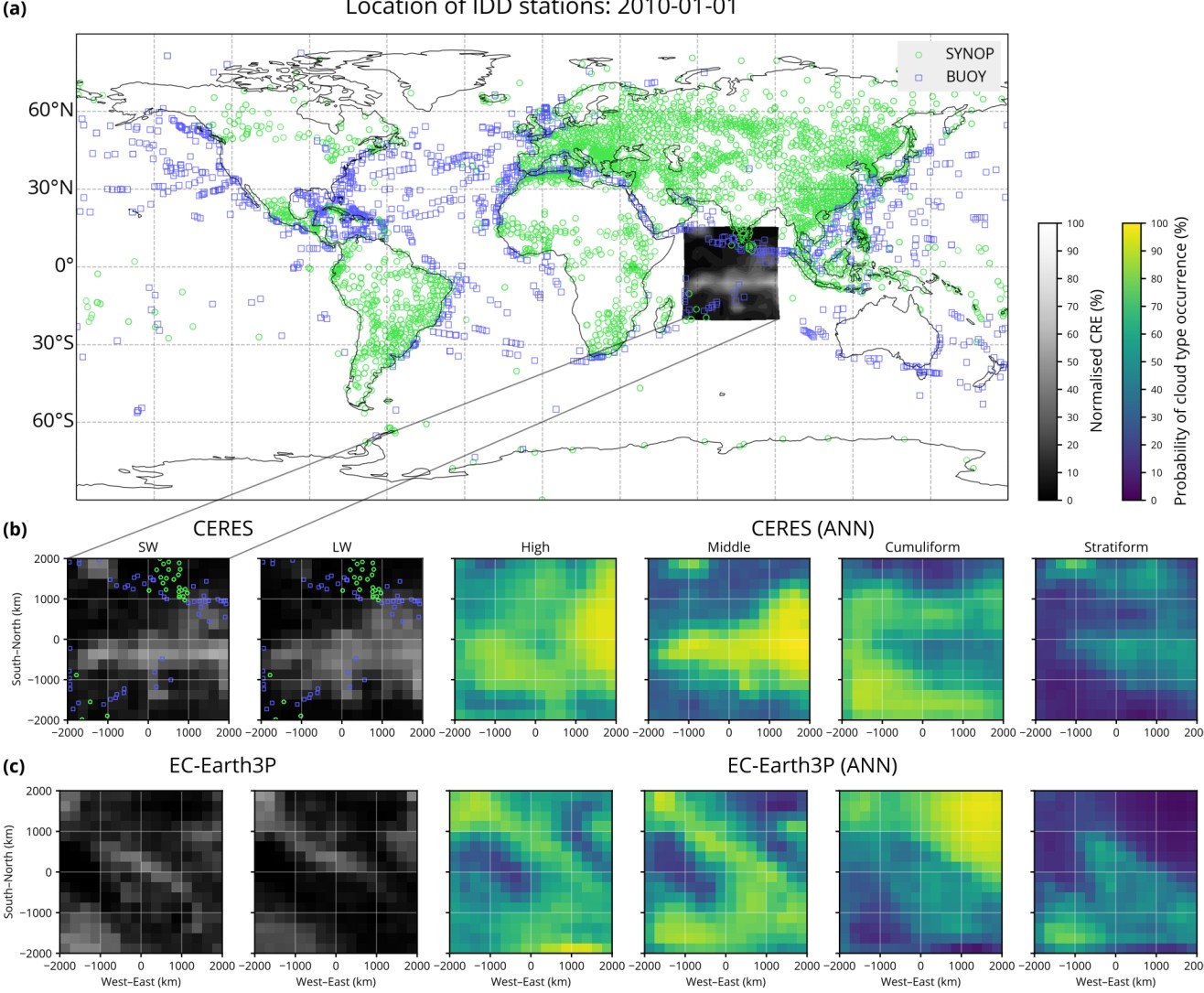

**Figure 1. (a)** A map showing the location of Internet Data Distribution (IDD) station reports containing cloud information on a single day. Shown is a sample of normalised top of atmosphere (TOA) shortwave radiation from CERES. **(b)** The sample as in (a), but shown re-projected in a local azimuthal equidistant projection. Shown is normalised shortwave (SW) and longwave (LW) cloud radiative effect (CRE), and the probability of cloud type occurrence calculated by the ANN for the classification into four cloud types. **(c)** A similar sample as in (b) but from the climate model EC-Earth3P (historical experiment) taken on the same day at a different location. This sample shows an unrelated cloud scene due to the fact that the model is free running. Note that the cloud types are not mutually exclusive, and therefore do not have to sum to 100%. Coastline data come from the public domain Global Self-consistent, Hierarchical, High-resolution Geography Database (Wessel and Smith, 1996, 2017).

is trained on a set of images either to label whole images (e.g. to identify if a certain object is present in the image), or to perform 'segmentation', where image pixels are classified as belonging to a certain object. Here, we used an ANN capable of quantifying the probability of presence of cloud genera in individual pixels of an image composed of shortwave and longwave channels coming from either satellite or model output. The spatial pattern and magnitude of shortwave (SW) and longwave (LW) radiation provides information about clouds, which can be used to train an ANN to classify clouds. The purpose was to quantify WMO cloud genera on the whole globe (rather than at individual station as available already in the IDD) and in model output. For practical purposes, in our analysis we grouped together multiple cloud genera to a smaller number of four and ten 'cloud types', in addition to using the full set of 27 WMO cloud genera. The ANN training phase consisted of supervised training on daily mean CERES satellite images, where for some pixels we knew the presence or absence of the cloud types based on one or more ground stations located within the pixel. The training of the output is done on these pixels. Because the number of stations and days of observations is relatively large, it was possible to train the ANN to quantify the probability of presence of cloud genera in any pixel of a satellite image or model output, and by extension on the whole globe.

The 'U-Net' ANN (Ronneberger et al., 2015) is a well-established type of ANN used for pixel-wise classification of images. The main feature of this network is its U-shaped sequence of steps of downsampling followed by upsampling, allowing it to learn patterns on different size scales and produce output of the same size as the input. This makes it suitable for our task of quantifying cloud type occurrence probability for each pixel in passive satellite images or an equivalent climate model output.

The schematic in Fig. 2 shows an outline of training and application ('prediction') of this ANN in our analysis. All input spatially-distributed satellite and model data were first resampled to 2.5° spatial resolution to ensure uniformity. In the training phase (Fig. 2a), samples of daily mean TOA SW and LW radiation from CERES were produced. A sample is an image of size $4000 \times 4000$ km and $16 \times 16$ pixels produced by projecting the daily mean TOA SW and LW radiation field in a local geographical projection at a random location on the globe. This step is necessary in order to produce input data which is spatially undistorted (as would be the case with unprojected global fields). Sampling at random locations ensures more robust training of the ANN, which could otherwise more easily be trained to recognise geographical locations as opposed to recognising cloud patterns irrespective of their location. Samples were paired with ground station cloud observations. Supervised training of the ANN was performed using these samples (20 per day, 4582 days in total). The training of cloud type occurrence probability, which is the output of the ANN, was only done for pixels where ground station data were available. In the application phase (Fig. 2b), samples from CERES or a model were produced and supplied to the ANN, which produced samples with quantified cloud type occurrence probability. These were then either merged to reconstruct a geographical distribution, or global means and trends were calculated from all samples. In detail, the ANN input were samples consisting of two channels of SW and LW radiation (256 values for each channel in 16 by 16 pixel samples), and the output were samples consisting of four, 10 or 27 channels (for classifications into four, 10 and 27 cloud types, respectively) of cloud occurrence probability corresponding to the cloud types (Fig. 2c). The classifications of four and 10 cloud types were created by grouping of the full set of 27 WMO cloud genera/species (as discussed earlier in Sect. 2.4). The advantage of using an ANN for cloud classification over more traditional methods such as partitioning the cloud top pressure–optical depth space (Rossow and Schiffer, 1991) is a true correspondence to human-identified cloud genera, its potential flexibility to identify more specific cloud genera/species, and the

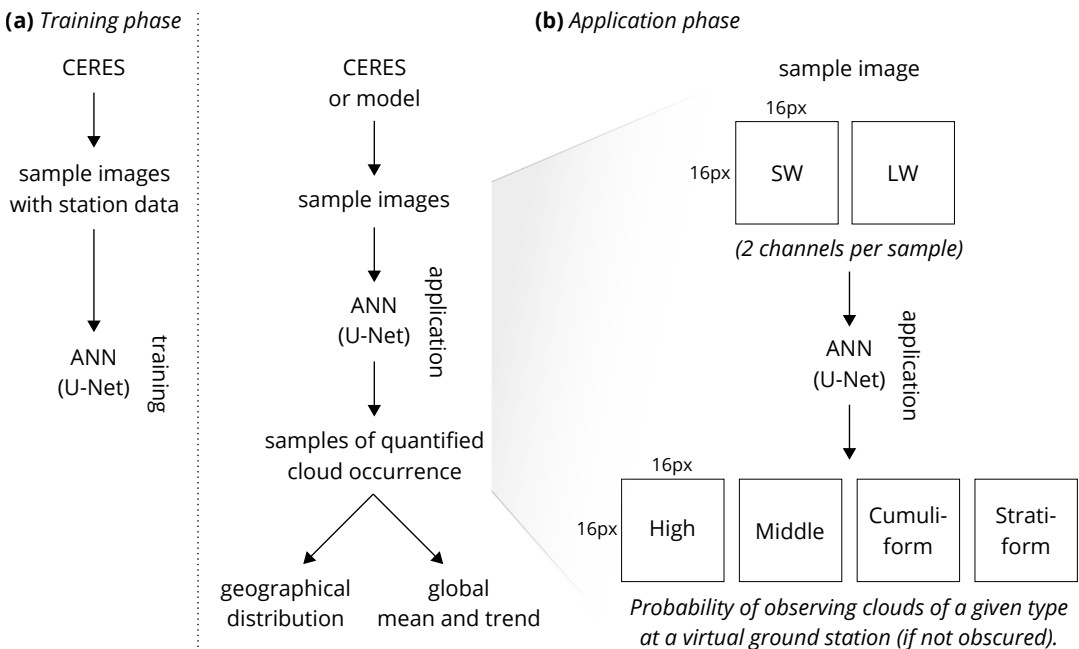

**Figure 2.** Schematic showing the principle of training and applying the ANN. **(a)** Sample images from CERES with reference ground station data (WMO cloud genera) are used in the *training phase*. **(b)** In the *application* ('prediction') *phase*, the ANN quantifies cloud type occurrence in sample images from CERES or a model. The ANN takes a sample image consisting of SW and LW channels and produces per-pixel quantification of probability of observing a given cloud type at a virtual ground station located in the pixel. The samples are 16×16 pixels in size and 4000×4000 km spatially.

ability to extend observations of human-identified cloud genera/species to regions not covered by ground stations or in time. The method outlined above allowed us to consistently quantify cloud genera/species occurrence in satellite observations and climate models and evaluate model biases.

## 220  3.2  Artificial neural network

TensorFlow is a machine learning framework for development of artificial neural networks (Abadi et al., 2016), supporting deep and convolutional neural networks. We used the Keras application programming interface (API) of TensorFlow (version 1.14), which provides a simple abstraction layer over TensorFlow, to define, train and apply a deep convolutional ANN on satellite and model data. The ANN was based on a network type called U-Net (Ronneberger et al., 2015), which produces 225 output on the same grid as the input. The input were 2-dimensional arrays of size 16×16 pixels for SW and LW radiation. The output were 2-dimensional arrays of the same size for each cloud type.

The ANN training was performed as follows, demonstrated schematically in Fig. 1 and 2. For each day, we generated 20 samples of 4000×4000 km and 16×16 pixels from CERES data, composed of two channels (shortwave and longwave radiation) projected in a local azimuthal equidistant projection centred at stochastically random locations uniformly distributed on the

globe (Fig. 1b). More precisely, the channels were calculated from daily mean TOA all-sky outgoing shortwave and longwave radiation (*rsut* and *rlut*, respectively), clear sky outgoing shortwave and longwave radiation (*rsutcs* and *rlutcs*, respectively) as (1) shortwave CRE normalised to the incoming solar radiation, (2) longwave CRE normalised to clear sky outgoing longwave radiation:

$$\text{CRE}_{\text{SW,norm.}} = (\text{rsut} - \text{rsutcs})/\text{rsdt}, \tag{1}$$

$$\text{CRE}_{\text{LW,norm.}} = (\text{rlutcs} - \text{rlut})/\text{rlutcs}. \tag{2}$$

The normalisation was done so that the values were mostly in the [0, 1] interval, which is a more suitable input to the ANN than non-normalised values. In the shortwave radiation, normalisation by incoming shortwave radiation was chosen so that the value represents the fraction of reflected incoming radiation due to clouds. In the longwave radiation, such normalisation is not possible, and normalisation by outgoing clear sky longwave radiation was performed instead.

In order to exclude locations with low solar insolation, where $\text{CRE}_{\text{SW,norm.}}$ might be ill-defined because of low values of the denominator, we excluded parts of $\text{CRE}_{\text{SW,norm.}}$ where incoming solar radiation was lower than 50 Wm$^{-2}$. A downside of this approach is that it may cause bias due to exclusion of wintertime polar regions. If a sample was missing any data points, it was excluded from the analysis. The shortwave and longwave channels were the input to the ANN training phase. The loss function of the ANN training was defined as the negative of log-likelihood of observing cloud types at ground stations. The
log-likelihood for each pixel and cloud type was:

$$l = n_{\text{positive}} \ln(p) + (n - n_{\text{positive}}) \ln(1 - p) \tag{3}$$

where $n_{\text{positive}}$ is the number of station records in the pixel which observed a given cloud type, $p$ is the probability of observing the cloud type predicted by the ANN, and $n$ is the total number of station records in the pixel with information about the cloud type. The total log-likelihood to be optimised was the sum of log-likelihood (as defined above) over all pixels in all samples
and all cloud types. Station records which reported clear sky were also included. In the optimisation process, in which the internal ANN coefficients were trained to predict the reference observations, the loss function equal to $-l$ was minimised. The optimisation process was run in iterations until the validation set loss function was not improved for three iterations.

In the application phase, 20 random samples per day were generated from CERES and model data. The ANN estimated the probability of cloud type occurrence for every pixel of each sample based on the input consisting of 16×16 pixel images of
SW and LW radiation calculated in the same way as in the training phase. From the samples we reconstructed geographical distributions and calculated global means or the probability of cloud type occurrence, i.e. the probability that a cloud type can be observed at a virtual ground station located in a given pixel.

### 3.3 Validation

Validation of the ANN was performed by comparing ANN predictions with the IDD in the validation years 2007, 2012 and
2017, which were not included in the training. In addition to validation of the ANN trained on all IDD station data available globally, we trained four ANNs by excluding IDD data over four geographical regions in the training:

- North Atlantic (15–45°N, 30–60°W)

- East Asia (30–60°N, 90–120°E)

- Oceania (15–45°S, 150–180°E)

- South America (0–30°S, 45–75°W)

The validation regions together with the number of available station reports in each grid cell are shown in Fig. S2. In addition, we trained four ANNs in which we excluded IDD data over one quarter of the globe in the training (north-west, north-east, south-east and south-west). With the abovementioned ANNs, we evaluated how the ANN performed when predicting over regions and times not included in the training dataset in comparison with the reference IDD data (Sect. 4.1).

To test if the validation regions are large enough to validate spatially uncorrelated locations, we analysed the temporal and spatial correlation in IDD station data (Fig. S3). The spatial correlation is approximately on the order of 1000 km, and temporal correlation is on the order several days.

For a validation of the ANN, we calculate the receiver operating characteristic (ROC) diagram (Sect. 4.1). An explanation of the diagram is given for example by Wilks (2019) in Chapter 9.4.6. The diagram shows sensitivity (the true positive rate)

and specificity (the true negative rate) of the prediction for a set of choices of thresholds for a positive prediction, represented on the diagram by points on a curve. The area under curve (AUC) is calculated by integrating the area under a curve, and can be intepreted as a goodness of the prediction.

### 3.4 Cloud top pressure–cloud optical depth evaluation

We calculated cloud top pressure–cloud optical depth histograms corresponding to the cloud types (Section 4.3). The his-

tograms were calculated from cloud top pressure and cloud visible optical depth variables in the CERES SYN1deg daily mean product over the time period of year 2003 to 2020 (inclusive) weighted by the daily cloud type occurrence calculated by the ANN. For each cloud type, the histogram density was calculated by iterating over all samples produced by the ANN and for every pixel incrementing the histogram bin corresponding to the pixel's cloud top pressure and cloud optical depth in the CERES product by the probability of the cloud type occurrence in the pixel calculated by the ANN. The set of samples did not

include random samples taken partially over polar night locations as explained in Sect. 3.2. The results shown in the histograms therefore do not include any information about polar regions during polar night.

### 3.5 Cloud properties by cloud type

We evaluated cloud fraction, cloud optical depth and cloud top pressure biases in models by ANN cloud type (Sect. 4.5). In CERES, they were taken from the SYN1deg (daily mean) product variables Adjusted Cloud Amount, Adjusted Cloud Optical

Depth and Observed Cloud Top Pressure, respectively. In CMIP, they were taken from the daily mean product variables cloud area fraction (*clt*), atmosphere optical thickness due to cloud (*cod*) and air pressure at cloud top (*pctisccp*), respectively. Cloud top pressure was taken from an International Satellite Cloud Climatology Project (ISCCP) simulator variable. In ERA5,

cloud fraction was taken from total cloud cover (*tcc*) (the other cloud properties were not available). In MERRA-2, the cloud properties were taken from total cloud area fraction (*CLDTOT*), in cloud optical thickness of all clouds (*TAUTOT*) and cloud

top pressure (*CLDPRS*), respectively. In CMIP, the free running *historical* experiment was used.

We calculated global mean of the cloud properties by cloud type as a weighted average of the above variables over the years 2003–2014 (models) and 2003–2020 (CERES), weighted by the product of the grid cell area and the cloud type occurrence probability determined by the ANN for the grid cell and day in the given model or CERES. The global mean did not include regions for which the ANN-determined cloud type occurrence probability was not available (polar regions in winter). We

compared global mean of each cloud property by cloud type in every model with CERES as an anomaly from CERES, i.e. model global mean over the time period 2003–2014 minus CERES global mean over the time period 2003–2020.

## 4    Results

### 4.1    Training and validation

We trained the ANN on CERES and IDD data in years 2004, 2005, 2009–2011, 2013–2016 and 2018–2020, with years 2007,

2012 and 2017 used as a validation dataset, representing 20% of the total number of years. The training was completed in 32, 40, and 38 iterations (for an ANN of 4, 10 and 27 cloud types, respectively), interrupted automatically once the validation loss function stopped improving for three iterations. The loss function during the training phase is in Fig. S1.

Figure 3 shows ANN validation results (the same but relative to the reference ANN is shown in Fig. S12). The geographical distribution of cloud type occurrence probability is determined by the ANN from samples of normalised CRE from CERES

in the validation years (2007, 2012 and 2017). A reference ANN trained on all stations data in the training years (Fig. 3a) is compared with four ANNs trained on station data in the training years excluding quarters of the globe at a time (Fig. 3b–e). In this way, we test if the ANN performs comparably to the reference ANN over regions where it had no station data to train on. It can be expected that excluding one quarter of the globe can result in a serious degradation of the prediction due to a lack of reference data for certain types of clouds common to particular geographical locations. Despite this limitation, the ANNs were

still able to reproduce large-scale patterns in the cloud type occurrence field as the ANN trained on all geographical locations. Some regions which were not reproduced well include the Himalayas (Fig. 3c) and high clouds over tropical west Pacific (Fig. 3 d1). The global area-weighted mean is similar between the reference ANN and the four validation ANNs, different most commonly by 0–4%. When calculated over the validation sectors only, the difference in area-weighted means is about 5% on average (Fig. S12). In summary, the ANN has some skill in extrapolating to geographical locations where no input data were

supplied in training, but some notable deviations in the mean exist.

Figure 4 shows the results of validation of the ANN against reference IDD data. Here a constant predictor (Fig. 4b) represents a reference predictor and results in RMSE of about 23% when comparing all-time means (years 2007, 2012 and 2017) between the predictor and the IDD, and 28% when comparing daily means between the predictor and the IDD. The ANN trained all years except the validation years (Fig. 4c, d) results in RMSE of about 17% and 23% in the comparison of all-time and daily

means, respectively. This represents about 28% and 19% fractional improvement of RMSE over the constant predictor. A

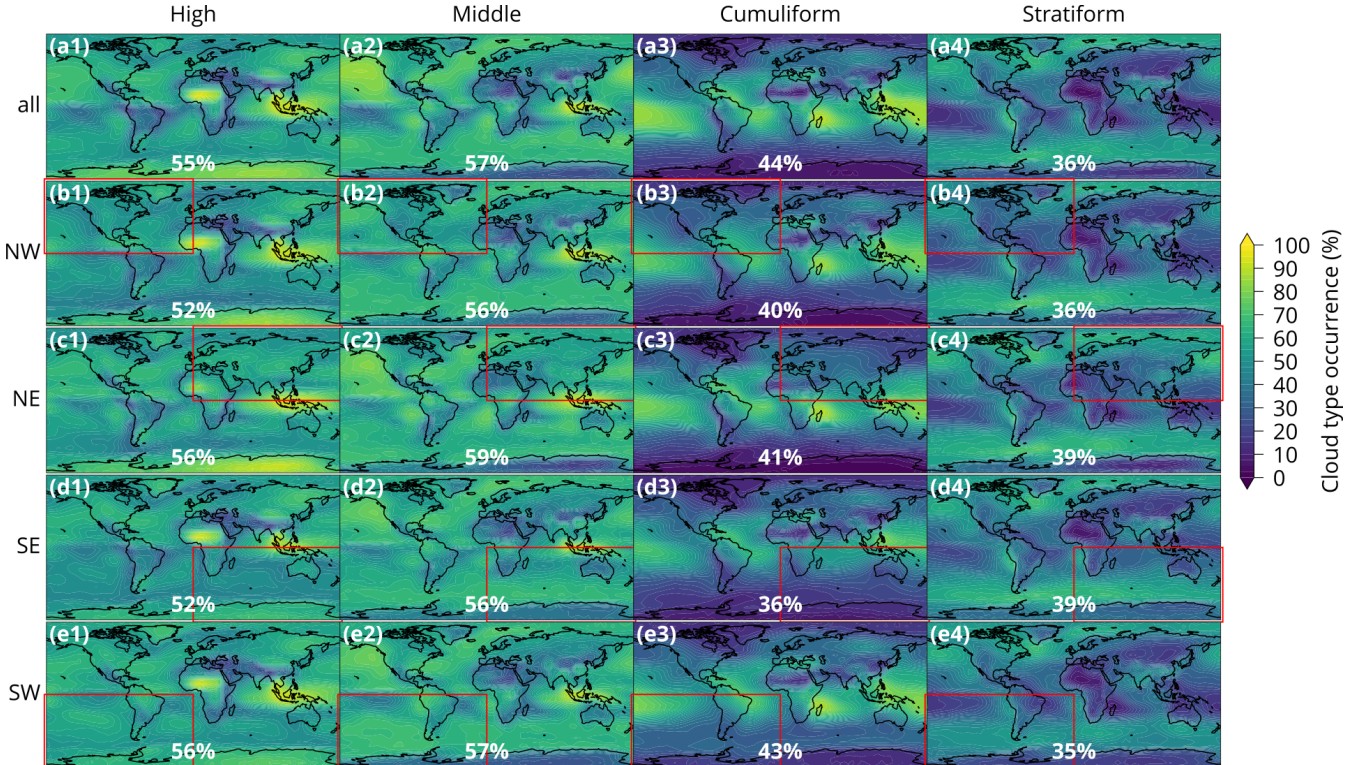

**Figure 3.** Geographical distribution of cloud type occurrence probability calculated by the artificial neural network (ANN) from CERES normalised cloud radiative effect (CRE) in the validation years 2007, 2012 and 2017. The plots show validation of the ANN by comparing a reference ANN with ANNs trained on station data excluding certain geographical regions. The row **(a)** is predicted by an ANN trained on all station data in the training years. The rows **(b–e)** are the same as (a), but predicted by ANNs trained on station data in the training years but excluding a quarter of the globe marked by a red rectangle (north-east (NE), north-west (NW), south-east (SE), south-west (SW)). The numbers in the lower centre of the plots show the area-weighted average cloud type occurrence probability over the whole globe.

composite of ANNs trained on all years except the validation years and excluding IDD data over four geographical regions (4e, f) results in RMSE of about 18% and 21% in the comparison of all-time and daily means, respectively. This represents about 22% and 25% fractional improvement of RMSE over the constant predictor. Note that the composite is created in a way that the cloud type occurrence probability in each geographical region is taken from the ANN which excluded IDD data over the region 330 in the training. In summary, the ANN shows substantial improvement when compared to a constant predictor when predicting in a time period not included in the training, as well as when predicting in regions not included in the training. However, the RMSE remains relatively large.

We note that the comparison above is quite strongly limited by the data availability. Included were all grid cell daily data with at least 15 reports of presence of absence of the cloud type, which can lead to an error on the scale of about 7% (100/15) 335 in the IDD reference. Moreover, only relatively sparse geographical locations had enough IDD data, mostly concentrated over

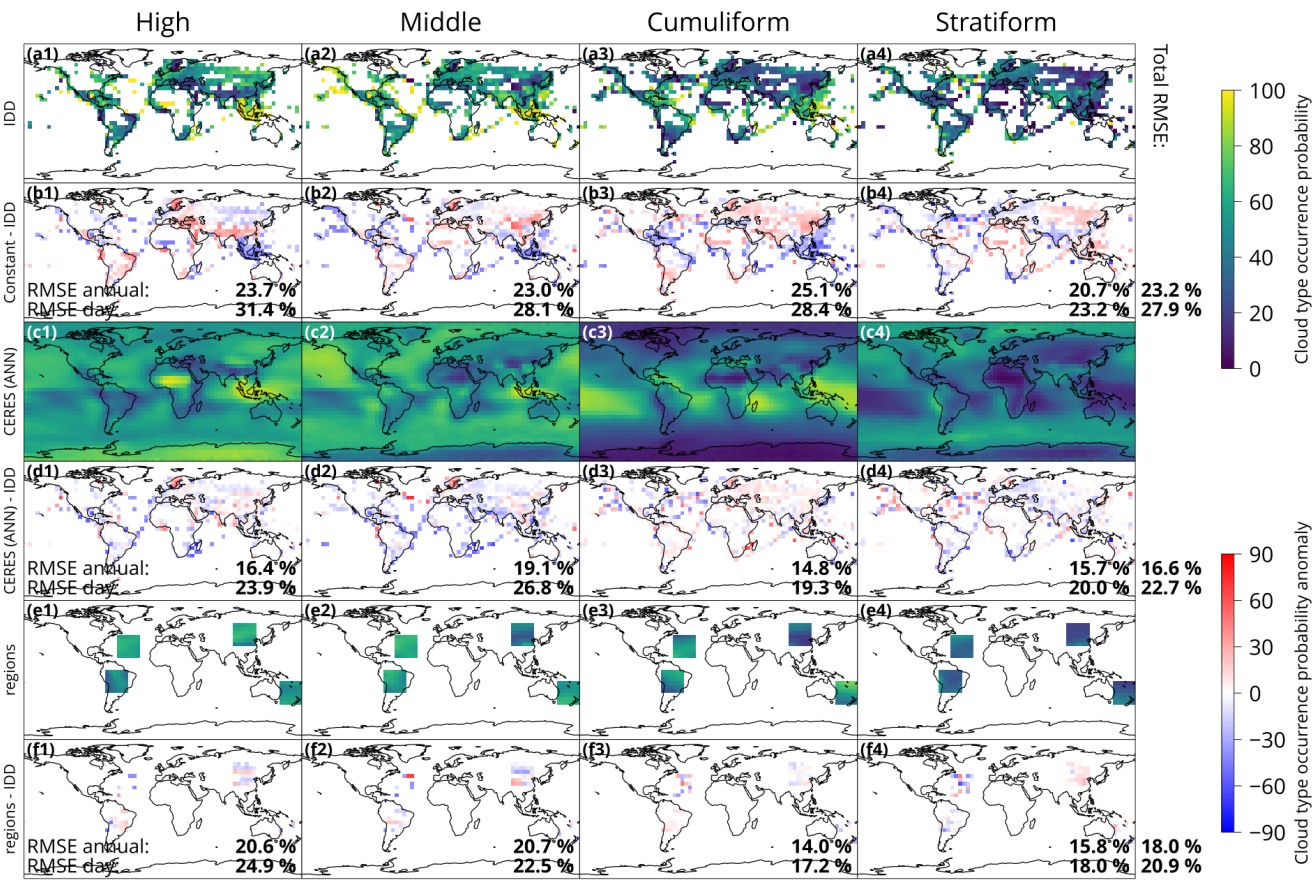

**Figure 4.** Validation of the artificial neural network (ANN) for four cloud types by comparison with Internet data distribution (IDD) on validation years 2007, 2012 and 2017. **(a)** Time mean of cloud type occurrence derived from IDD on a $5 \times 5°$ grid. Included are only grid cells where at least 70% of days with 15 or more station reports containing the cloud type information are available. **(b)** A constant predictor relative to IDD (a), which is one which predicts cloud type occurrence probability equal to the global spatiotemporal mean of the cloud type occurrence probability calculated over the training time period (2004–2020, excluding the validation years). **(c)** Time mean of cloud type occurrence probability predicted by the ANN on the validation years. **(d)** The same as (c), but relative to IDD (a). **(e)** As (c), but a composition of four separate ANNs trained on validation regions (North Atlantic, East Asia, Oceania and South America), where each shown region comes from the ANN trained on station data over the training period excluding the given region. **(f)** The same as (e), but relative to IDD (a). Shown in the panels is root mean square error (RMSE) relative to IDD calculated by comparing two time means over all of the validation years ('RMSE annual'), and daily means ('RMSE daily'). On the right, the total RMSE is shown, calculated from the four cloud type root mean square errors ($\text{RMSE}_i$) as $(1/4 \sum_{i=1}^{4} \text{RMSE}_i^2)^{1/2}$.

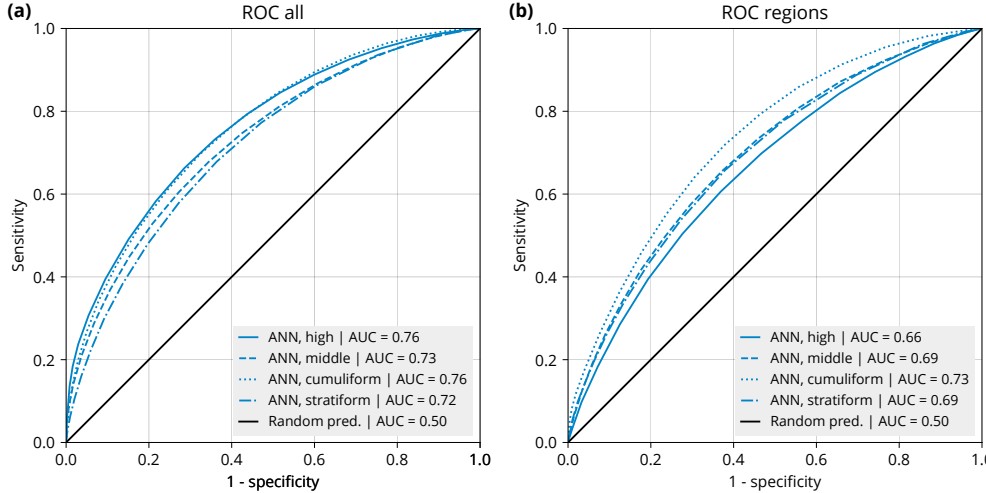

**Figure 5.** Receiver operating characteristic (ROC) diagram calculated for **(a)** an artificial neural network (ANN) for the four cloud types trained on all training years (2003–2020, except the validation years 2007, 2012 and 2017) on all data available globally, evaluated against station reports from the Internet Data Distribution (IDD) in the validation years, **(b)** a composite of four ANNs trained on all training years on all data available globally except for each of the four validation regions (North Atlantic, East Asia, Oceania and South America), evaluated against station reports from the IDD in the respective region in the validation years. Shown is also the ROC of a random predictor. Sensitivity is the true positive rate (probability of a positive prediction if positive in reality), also called the hit rate. Specificity is the true negative rate (probability of a negative prediction if negative in reality). '1 - specificty' is also called the false alarm rate. Area under curve (AUC) of the ROCs is in the label.

land. In the supplement we include equivalent figures to Fig. 3 and 4 but for 10 and 27 cloud types. We note, however, that the error due to the number of available IDD station reports per grid cell per day (as mentioned above) may be a large part of the RMSE in these figure, especially for the 27 cloud types (a large number of reports needs to be available in a grid cell to accurately quantify the occurrence of rare cloud types).

Figure 5 shows the ROC diagram calculated for a comparison between IDD and an ANN trained on all IDD data in the training years (Fig. 5a), and a composite of four ANNs trained on all IDD data in the training years excluding the four validation regions (Fig. 5b). Here, the reference IDD data include all station reports in the validation years 2007, 2012 and 2017. All cloud types were predicted with similar accuracy in terms of the ROC and AUC, with an average AUC of 0.74 in the all-data ANN and 0.69 in the regions composite. As expected, the composite performs more poorly than the all-data case. The

ROC of the ANNs can be interpreted as a substantial improvement over a random predictor.

    In summary, the ANN has a good performance when compared to trivial predictors, but more substantial errors are present on daily scales. The ANN shows relatively good ability to extrapolate to regions not included in the training. Regions which do not have an analogue in the rest of globe, such as the Himalayas and tropical west Pacific, are crucial for the training, and without training data over the regions the ANN does not perform well over the regions. The validation results show that

the ANN has the ability to reproduce large-scale patterns successfully, and therefore can be used in large-scale analysis, but it might not be accurate enough to capture smaller-scale and daily-scale variations. This might be due to the rather severe limitation imposed by the low spatial resolution of the input data. The presented validation is, however, itself limited by sparse spatial availability of the IDD data.

## 4.2 Geographical biases

Fig. 6 and 7 show the geographical distribution of cloud types in CERES, IDD, CMIP models and the reanalyses (ERA5 and MERRA-2) for the four cloud types (analogous plots for 10 and 27 cloud types are available in the supplement). The IDD row represents an observational reference calculated independently from the ANN (although the ANN is originally trained on this dataset), while rows corresponding to CERES and the CMIP models are calculated by the ANN. The high cloud type is characterised by a peak over the western tropical Pacific and tropical Africa, corresponding to peaks in the IDD. A peak over

northern Asia is, however, more muted in the ANN. The middle cloud type peaks over northern Pacific and western tropical Pacific, and has a minimum over central Africa. Northern Pacific is not sampled well in the IDD, but a peak over western tropical Pacific and a minimum over central Africa are also present in the raw dataset. The cumuliform cloud type is strongly concentrated in tropical marine regions over tropical Pacific and Atlantic and Madagascar and minima over polar regions and tropical Africa. This is also present in the IDD. The stratiform cloud type has maxima over polar marine regions and on the

west coast of South America, and minima over tropical Pacific, Madagascar and tropical Africa. The maxima in the IDD are co-located but stronger, while the minima are co-located and of similar magnitude.

Model biases are most strongly characterised by a negative bias in the cumuliform cloud type over marine tropical regions and a positive bias in the stratiform cloud type over the same regions, especially in models with lower ECS (indicated in Fig. 6 below the model name). Bias in the high and middle cloud type is more geographically varied. Models with higher ECS tend

to have the opposite bias – positive bias in the cumuliform cloud type and negative bias in the stratiform cloud type globally. Notably, models with lower ECS generally have higher biases than models with higher ECS. Of models with ECS below 4 K (9 models), all but two have total RMSE greater than or equal 8%. Of models with ECS above 4 K (9 models), all have RMSE lower than 8%. The two reanalyses (ERA5 and MERRA-2) have relatively low biases compared to the CMIP models. ERA5 has the lowest total RMSE of all models at 3.6%.

Models which are closely related in their code (CNRM-*; ERA5 and EC-Earth3P; HadGEM3-* and UKESM1-0-LL; INM-*; IPSL-*; MPI-*) performed similarly in terms of geographical distribution and magnitude of biases. This means that the ANN method is robust with respect to model resolution, and also that the groups of related models represent clouds very similarly, presumably because this is to a large extent determined by cloud parametrisations in the atmospheric component of the model, without much sensitivity to resolution.

## 380 4.3 Optical properties and vertical distribution

From the ANN-labelled samples we calculated joint histograms of cloud optical depth and cloud top pressure (Fig. 8). This type of histograms relates to previous work on cloud classification of Rossow and Schiffer (1991), Rossow and Schiffer (1999),

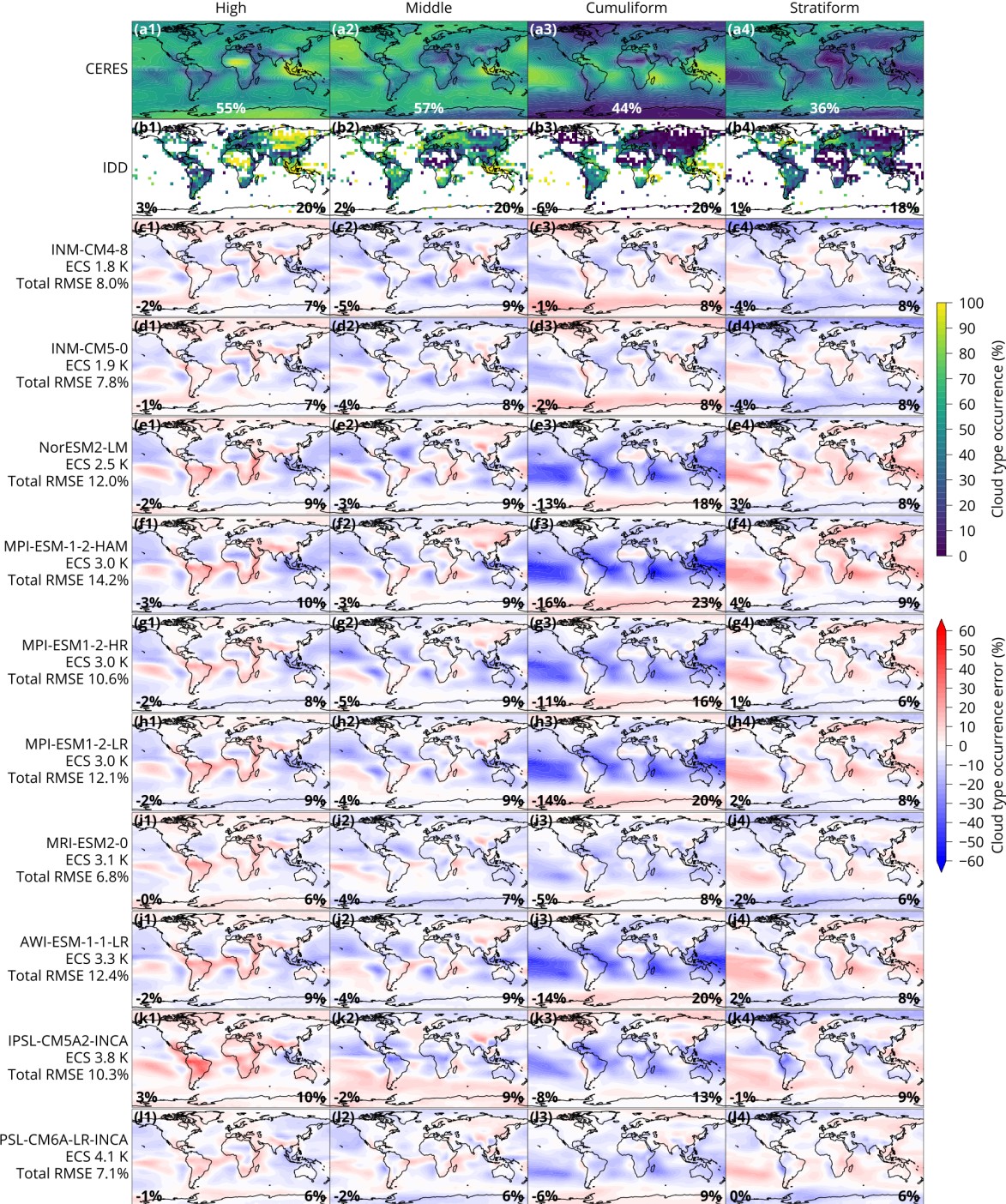

**Figure 6.** Geographical distribution of cloud type occurrence derived by applying the ANN on retrieved CERES satellite data in years 2003–2020 **(a1–a4)**, directly from the IDD (year 2010) **(b1–b4)**, and by applying the ANN on model output of the historical experiment of CMIP6 and reanalyses in years 2003–2014 relative to CERES **(c–k)**. In the lower centre of the CERES plots is the geographical mean occurrence of the cloud type. In the lower left is the mean error and in the lower right is the RMSE. Models are sorted by their ECS from lowest to highest. Coastline data come from the public domain Global Self-consistent, Hierarchical, High-resolution Geography Database (Wessel and Smith, 1996, 2017).

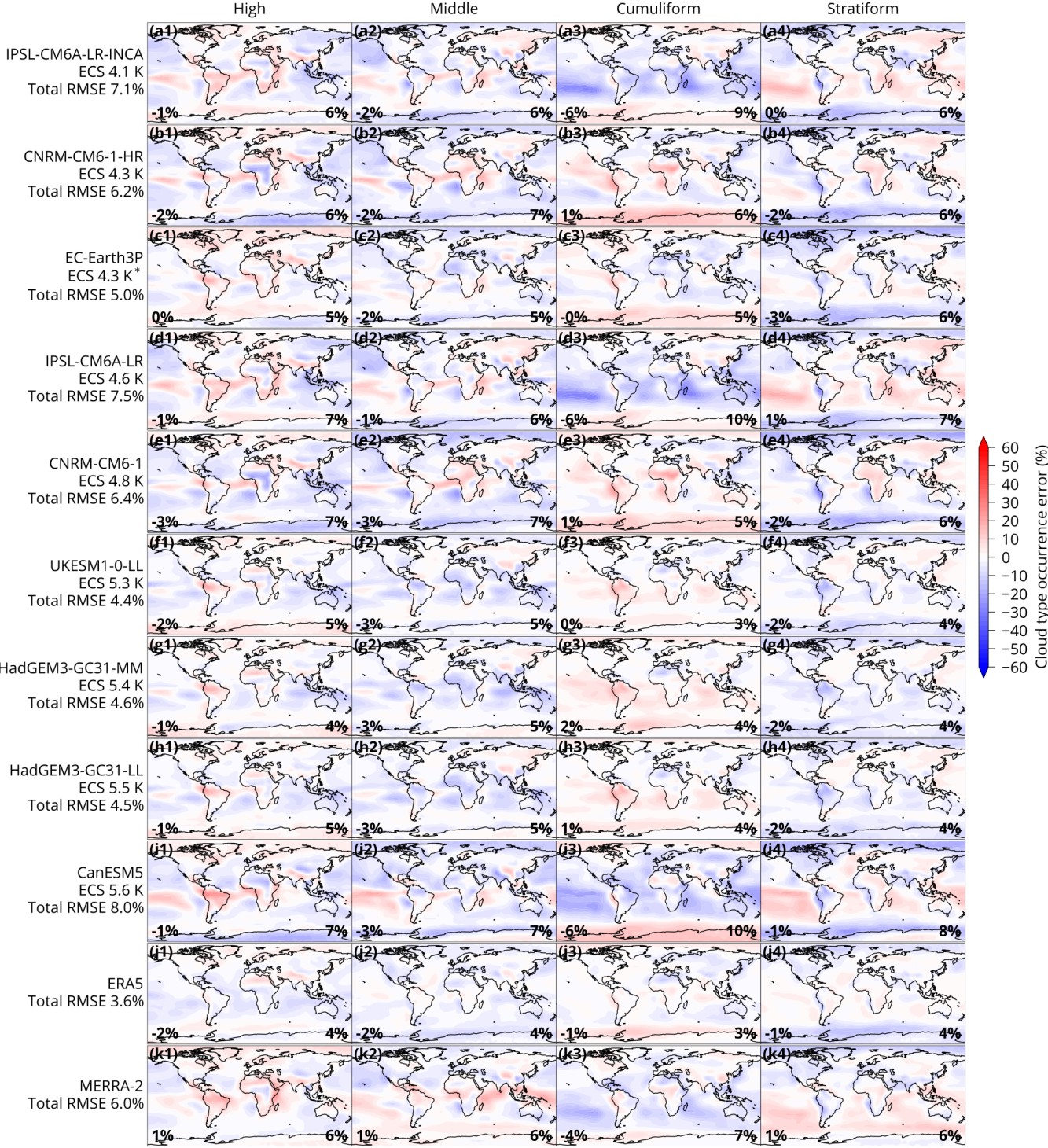

**Figure 7.** Fig. 6 continued. *For some models, ECS was not available and was taken from a related available model (see Table 1).

Hahn et al. (2001), Oreopoulos et al. (2016), Schuddeboom et al. (2018) and others. The diagrams in Fig. 8 show cloud type occurrence binned by cloud optical depth and cloud top pressure for the four types as a difference from the mean of the four types. The high cloud type difference from the mean is characterised by a maximum occurrence at low pressure (300–700 hPa) and low optical depth (below 5) (Fig. 8b). The middle cloud type difference from the mean is characterised by high optical depth (above 2) between 200 and 800 hPa (Fig. 8c). The cumuliform and stratiform types have greater deviation from the mean than the high and middle types. The cumuliform cloud type difference from the mean has a maximum in optically thin clouds (below 4) between 400 and 1000 hPa (Fig. 8d). The stratiform cloud type is almost the inverse of the cumuliform type (Fig. 8e), characterised by mid-to-high optical depth (above 2) and low-to-mid altitude (below 400 hPa, peaking at about 700 hPa). Collectively, the cloud types span discrete regions in four sectors of the diagram, which indicates that they partition the cloud optical depth–cloud top pressure space quite well with little overlap. This means that the classification method distinguishes well between types of clouds in terms of their cloud optical depth and cloud top pressure. This analysis also shows that the stratiform and middle cloud types as identified by the ANN are generally more opaque in terms of cloud optical depth than the cumuliform and high cloud types.

To compare with a more traditional classification by Rossow and Schiffer (1991), we present a diagram corresponding to their International Satellite Cloud Climatology Project (ISCCP) classification (Fig. 8f). Correspondence of the cumuliform and stratiform types between the ANN and the ISCCP classification is quite good, except for deep convection (classified as the cumuliform type in the ISCCP diagram) of highly opaque (optical depth above 20) high clouds (above 400 hPa). The correspondence is less good for the high cloud type, where the ISCCP class starts at higher altitude (400 hPa vs. 700 hPa) and is also more opaque (optical depth up to 20 vs. 5). The correspondence of the middle cloud type is relatively poor – the ISCCP middle clouds are located at lower altitude (500 hPa vs. 400 hPa) and have lower optical depth (down to 0 vs. down to 2). Therefore, the correspondence between our classification and the ISCCP classification of Rossow and Schiffer (1991) is mixed, with good correspondence of the low cloud types, but disparities in the middle and high cloud types. This may be due to the fact that our method is based on ground-based cloud observations, which are often not capable of identifying high and mid-level clouds. It can be expected that discrimination of high and mid-level clouds by the ANN is not as good as that of low clouds.

In addition to the comparison with the ISCCP classification above, in Appendix B we present a comparison with cloud clusters derived using self-organising maps (SOM) of McDonald and Parsons (2018) and Schuddeboom et al. (2018).

## 4.4 Cloud type global climatology and change with global mean near-surface temperature

We analysed global mean cloud type occurrence in the CMIP historical experiment models and the reanalyses, and change with respect to GMST in the abrupt-4xCO2 experiment. The abrupt-4xCO2 experiment was chosen because it (1) is commonly used for the determination of ECS and cloud feedback, (2) provides a strong forcing by greenhouse gases and therefore a strong signal in cloud change due to increasing GMST, (3) a large number of models provide the necessary data in this experiment in the CMIP5 and CMIP6 archives. As shown in Fig. 9a comparing model global mean cloud type occurrence relative to CERES, the models exhibit a broad range of biases, but with many similarities. Underestimation of the cumuliform cloud by up to

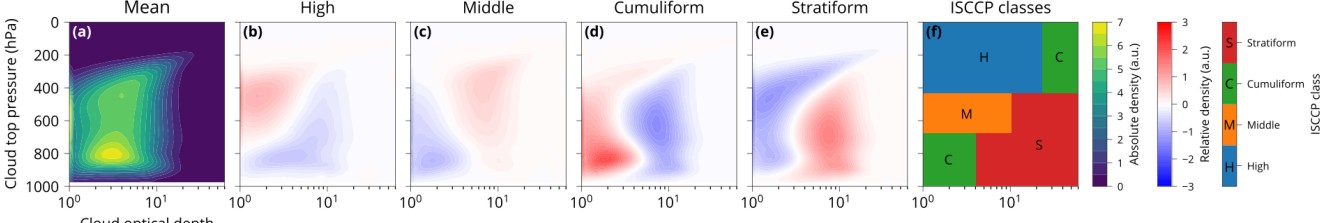

**Figure 8.** Histogram of cloud optical depth and cloud top pressure of the cloud types derived from CERES (2003–2020) by applying the ANN. **(a)** Mean of the four cloud types. **(b–e)** Difference from the mean for the classification of four cloud types. **(f)** ISCCP classification (Rossow and Schiffer, 1991).

19% is common, as well as smaller underestimation of the middle cloud type (up to 6%) and high cloud type (up to 3%). Overestimation of the stratiform type up to 5% was present in a smaller number of models. Progression from large biases to low biases in the cumuliform and middle cloud type with increasing model ECS is quite notable, with the exception of INM-*

and to a lesser extent IPSL-CM6A-LR and CanESM5. In particular, most models with lower ECS (< 4 K) tend to underestimate the cumuliform type and some also overestimate the stratiform type, while models with higher ECS (> 4 K) to a smaller degree tend to underestimate the stratiform type and some overestimate the cumuliform type. The reanalyses (ERA5 and MERRA-2) have some of the best agreement with CERES compared to the CMIP models.

We also analysed cloud type occurrence change with respect to GMST, defined as the slope of linear regression of cloud

type occurrence as a function of GMST in units of $\% \, K^{-1}$, shown in Fig. 9b. It was calculated from year 1 to 100 of the CMIP abrupt-4xCO2 experiment. Some models do not provide all years in this time period. These models are MPI-ESM-LR, for which we used years 1850–1869 and 1970–1989, as in the *time* variable of the product files, and MRI-CGCM3, for which we used years 1851–1870 and 1971–1990. These years do not correspond to real years, but rather an arbitrary time period starting with 1850 used for the abrupt-4xCO2 experiment in these models. This comparison lacks a reliable observational reference

because the CERES record is too short to accurately determine the slope of the regression. The abrupt-4xCO2 experiment is also not directly comparable to reality due to the different $CO_2$ and aerosol forcing. The models exhibit a broad range of values with few common trends. Models with lower ECS (< 4 K) tended to simulate increasing stratiform and middle cloud type and decreasing cumuliform type, while models with higher ECS (> 4 K) tended to simulate decreasing stratiform and middle cloud type and increasing cumuliform type. This behaviour is consistent with the warming effect of stratiform and middle clouds. In

our analysis, the cumuliform cloud type has lower optical depth than the stratiform and middle cloud types (Sect. 4.3), and in this sense one would expect more warming if the cumuliform cloud type is replaced with the stratiform cloud type (and vice versa). The corresponding geographical distribution plots are provided in Fig. S7 and S8.

Figure S4–S6 show the same as Fig. 9, but for a classification into 10 and 27 cloud types, respectively. We note, however, the classification into 27 cloud types should be considered with caution due to the fact that association between the low-resolution

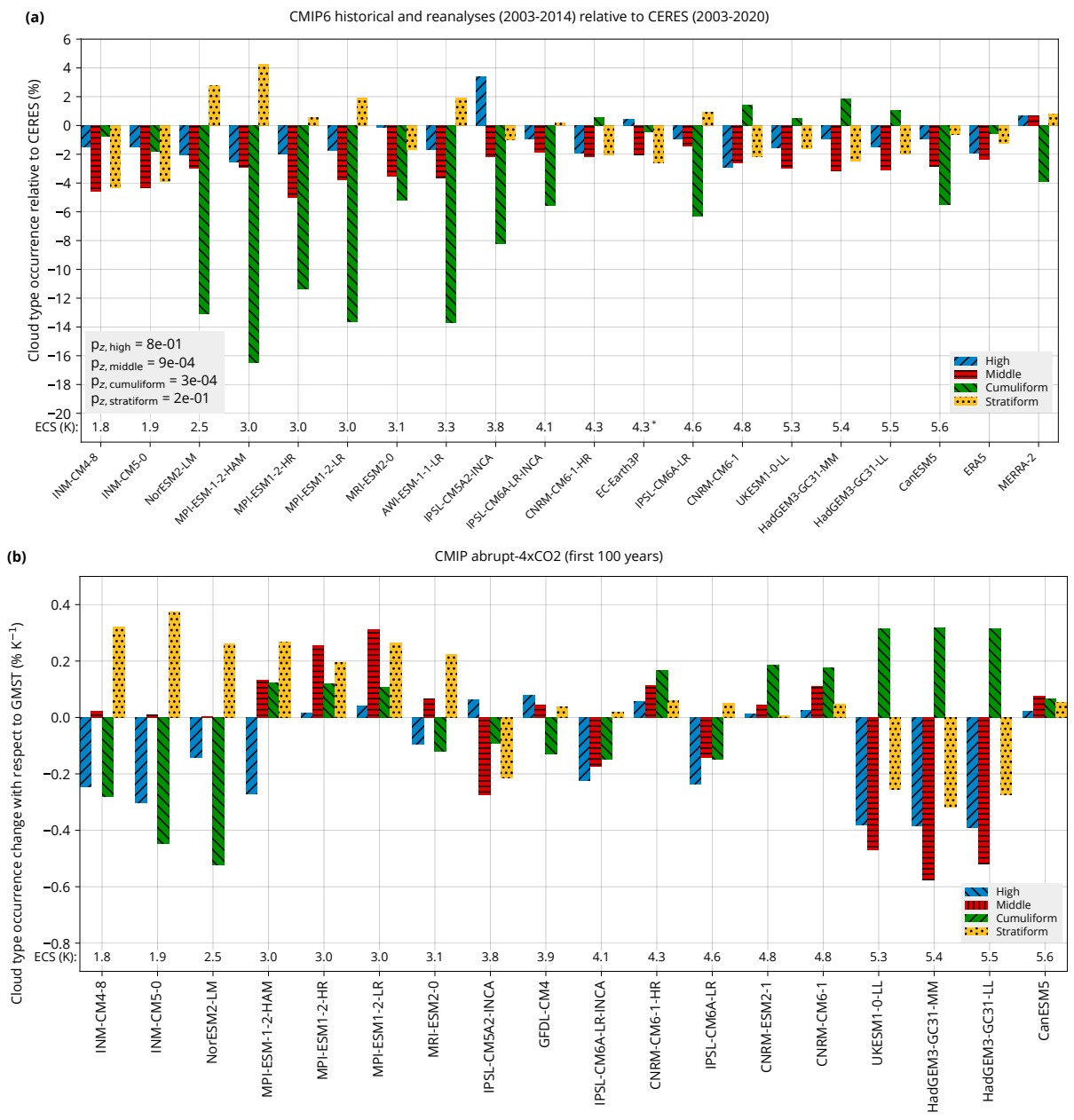

**Figure 9. (a)** Global mean cloud type occurrence in CMIP6 models and reanalyses relative to CERES in the historical experiment. Shown is also the p-value of a z-test for the difference of means of two groups of CMIP models with ECS below and above 4 K (the mid-point ECS in the range of the analysed models). **(b)** Global mean of cloud type occurrence change with respect to global mean near-surface air temperature (GMST) in CMIP5 and CMIP6, calculated by linear regression. Models are sorted by their equilibrium climate sensitivity (ECS). *For some models, ECS was not available and was taken from a related available model (see Table 1).

TOA radiation and the cloud type occurrence is inferential, and the individual cloud genera/species in general cannot be directly observed in the radiation fields.

## 4.5   Cloud properties by cloud type

We analysed cloud properties categorised by the ANN cloud types. Figure 10 shows cloud fraction, cloud top pressure and cloud optical depth by cloud type in all CMIP models and reanalyses as an anomaly from CERES, for which the data were available.

The cloud properties display a relative large degree of similarity irrespective of the ANN cloud type, especially in cloud top pressure and cloud optical depth. Cloud top pressure of the high cloud type had greater negative bias than the other cloud types in the four CMIP models analysed. This is also notable due to the fact that this represents a larger relative error in pressure (and larger difference in height) for high clouds than for low and mid-level clouds. Cloud fraction was underestimated relative to CERES in most models and reanalyses, with the following exceptions. The INM models overestimated cloud fraction,

except for the stratiform cloud type. The HadGEM/UKESM and CNRM-CM6-1-HR models were relatively close to CERES compared to the rest of the models and reanalyses. This coincides with the outlying properties of INM in the rest of our analysis (Sect. 4.6), as well as good performance of HadGEM/UKESM in representing the ANN cloud type occurrence (Sect. 4.2). Cloud top pressure was underestimated in all models and reanalyses (but only five were available in this comparison). Cloud optical depth was overestimated in the four analysed models and reanalyses, but in the HadGEM/UKESM models with

a much smaller magnitude than the other two models and reanalyses.

In summary, the results point to a general too few, too bright cloud problem identified in previous studies (e.g. Nam et al. (2012); Klein et al. (2013); Engström et al. (2015); Wall et al. (2017); Bender et al. (2017); Kuma et al. (2020); Konsta et al. (2022)) and higher altitude of clouds in the models and reanalyses than in the satellite observations. There was no clear dependence of cloud fraction on the model ECS, unlike the dependence of cloud type occurrence probability on model

ECS (Sect. 4.4). The analysis of the cloud properties is limited by several caveats, such as that the cloud properties are not necessarily reliably comparable between models and observations without the use of an appropriate instrument simulator. We used non-simulator cloud fraction from the CMIP models because of the wider availability of data than a corresponding simulator variable. Cloud top pressure was derived from a simulator, but for a different satellite dataset (ISCCP). Cloud optical depth in CMIP was only available as a non-simulator based variable. We also note that this analysis of cloud properties only

applies spatiotemporally to a domain covered by the cloud type occurrence evaluation, which excludes polar regions in winter. Therefore, they are spatiotemporally biased to non-polar regions and non-winter seasons.

## 4.6   Climate sensitivity

We analysed how cloud type occurrence change with respect to GMST relates to climate sensitivity. ERA5 and MERRA-2 are excluded from the analysis in this section because climate sensitivity and feedbacks are not estimated for reanalyses. Fig. 11

shows a linear regression of ECS as a function of model's cloud type occurrence change with respect to GMST. The relation of ECS with the stratiform cloud type is the strongest (probability of the null model representing no linear relationship in the data $P(M_0) = 3 \times 10^{-4}$; see Appendix A), with the cumuliform type slightly less strong ($P(M_0) = 2 \times 10^{-3}$), and with the middle

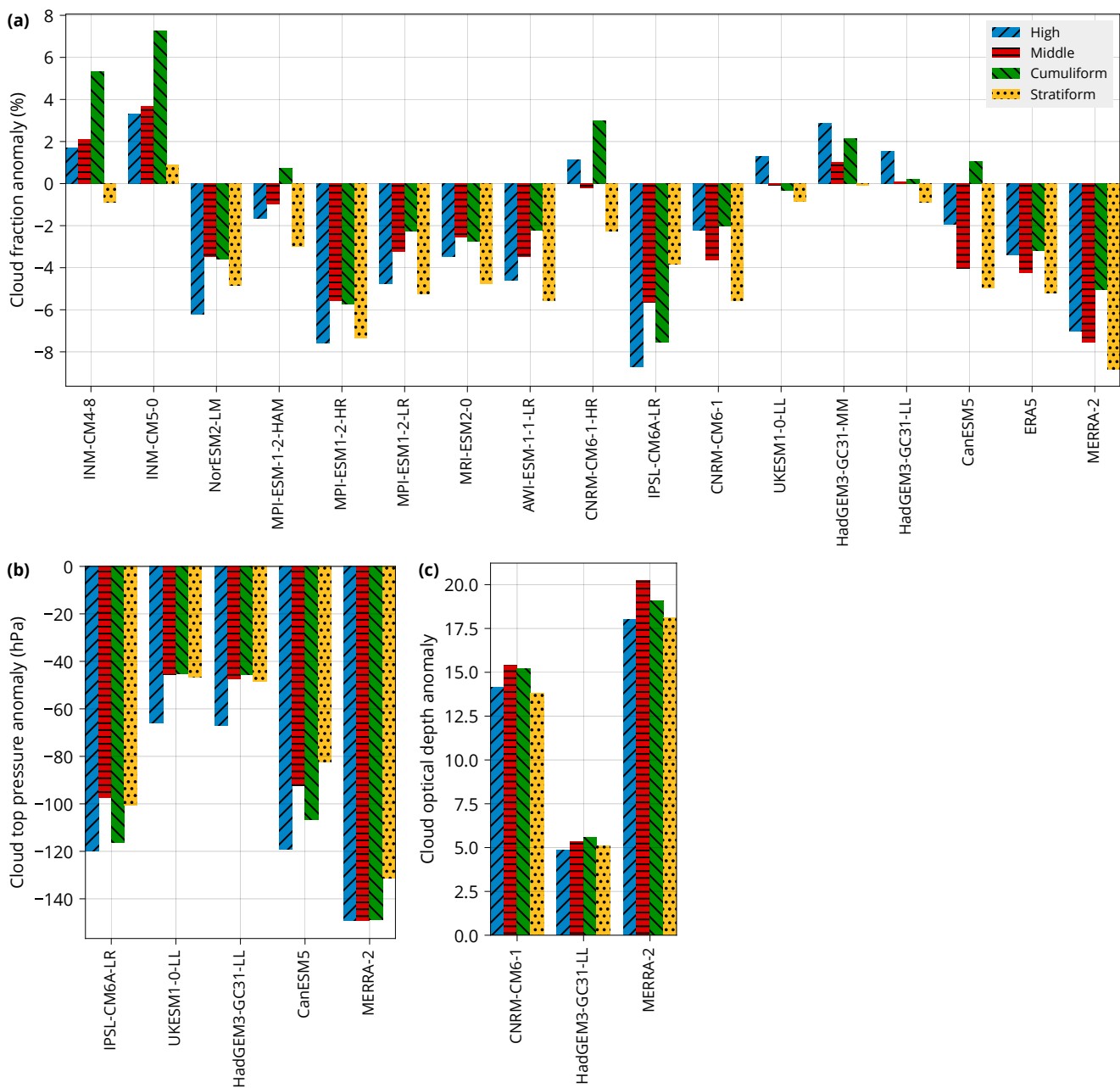

**Figure 10.** Cloud properties in CMIP models and reanalyses by cloud type relative to CERES. The bar charts show area-weighted global mean of cloud properties calculated over the domain where cloud types are determined by the ANN (all locations except polar regions in winter). Cloud properties shown are **(a)** cloud fraction, **(b)** cloud top pressure, and **(c)** cloud optical depth. In the CMIP models, cloud top pressure is from the ISCCP simulator. All other cloud properties are from non-simulator variables. For each cloud type, the mean is calculated from daily data by weighting values by the cloud type occurrence determined by the ANN for the particular model or CERES in every grid cell and time step. The model and reanalysis data are for years 2003–2014, and the CERES data are for years 2003–2020. The models are sorted by their ECS from the lowest to highest.

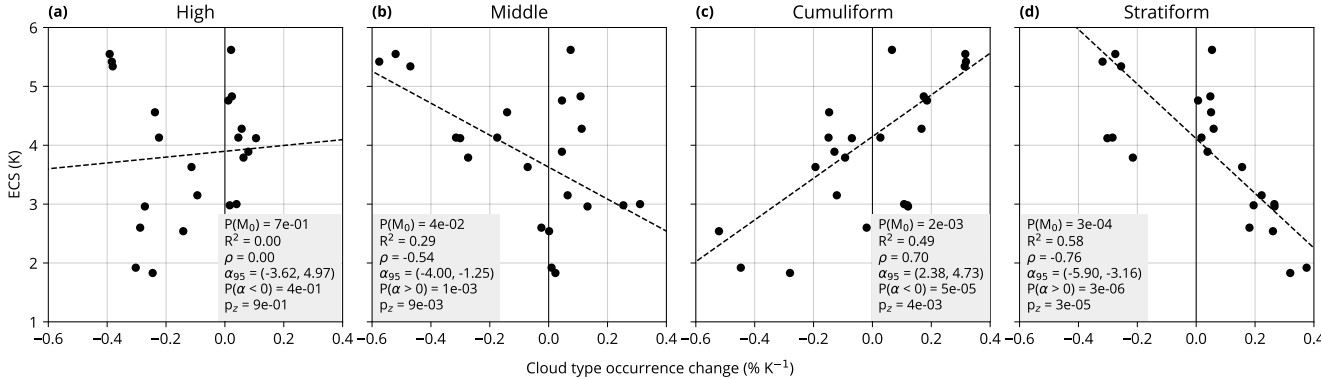

**Figure 11.** Dependence of model equilibrium climate sensitivity (ECS) on the cloud type occurrence change with respect to GMST. Confidence bands represent 68% range. Linear regression is calculated using Bayesian simulation assuming Cauchy error distribution (Appendix A). Shown is also the probability of the null hypothesis model ($P(M_0)$) (explained in Sect. 4.6), the coefficient of determination ($R^2$), correlation coefficient ($\rho$), 95% confidence interval of the slope ($\alpha_{95}$) of the linear regression, probability that the slope is smaller or greater than zero ($P(\alpha < 0)$ and $P(\alpha > 0)$), and the p-value of a z-test ($p_z$) for the difference of means of two groups of models in the bottom 50% and top 50% of ECS. *For some models, ECS was not available and was taken from a related available model (see Table 1).

cloud type relatively strong ($P(M_0) \approx 0.04$). Relation with the high cloud type was not statistically identifiable (probability below 5%). This is also confirmed by a z-test for the difference of means of two groups of models with ECS below and above

4 K (the mid-point in the ECS range of the analysed models), with a p-value of $3 \times 10^{-5}$, $4 \times 10^{-3}$ and $9 \times 10^{-3}$ for the stratiform, cumuliform and middle cloud type, respectively. Higher ECS is associated with decreasing stratiform and middle cloud types and increasing cumuliform cloud type with increasing GMST. This may be physically explained by the fact that the cumuliform cloud type has low optical depth compared to the stratiform type (Fig. 8), and therefore if a model simulates a transition from stratiform to cumuliform clouds with increasing GMST, radiative forcing due to cloud is increased. We note

that for the Bayesian statistical analysis results (probability of the null model) we used priors for the null and alternative models both equal to 0.5 (Appendix A).

Cloud type change with respect to GMST is too uncertain in the observational reference (CERES) to be be useful for quantifying the accuracy of models in the representation of this value. The abrupt-4xCO2 experiment assessed here is also not directly comparable to reality. However, we can link present-day cloud biases to climate sensitivity. In Fig. 12 we show that

the total RMSE of cloud type occurrence (calculated for the 27 cloud types) is linearly related to the model ECS ($P(M_0) = 2 \times 10^{-3}$), TCR ($P(M_0) = 9 \times 10^{-3}$) and cloud feedback ($P(M_0) = 1 \times 10^{-2}$). Models with the lowest RMSE tend to have the largest ECS, TCR and cloud feedback, while models with the highest RMSE tend to have the lowest ECS, TCR and cloud feedback. There are, however, several outliers such as INM-* with mid-range RMSE and the lowest ECS and TCR of all models in the ensemble, and CanESM5 with mid-range RMSE and the highest ECS. The relationship could also be artificially

strong due to cross-correlation between related models. If only one model of each model family is retained (10 out of 18),

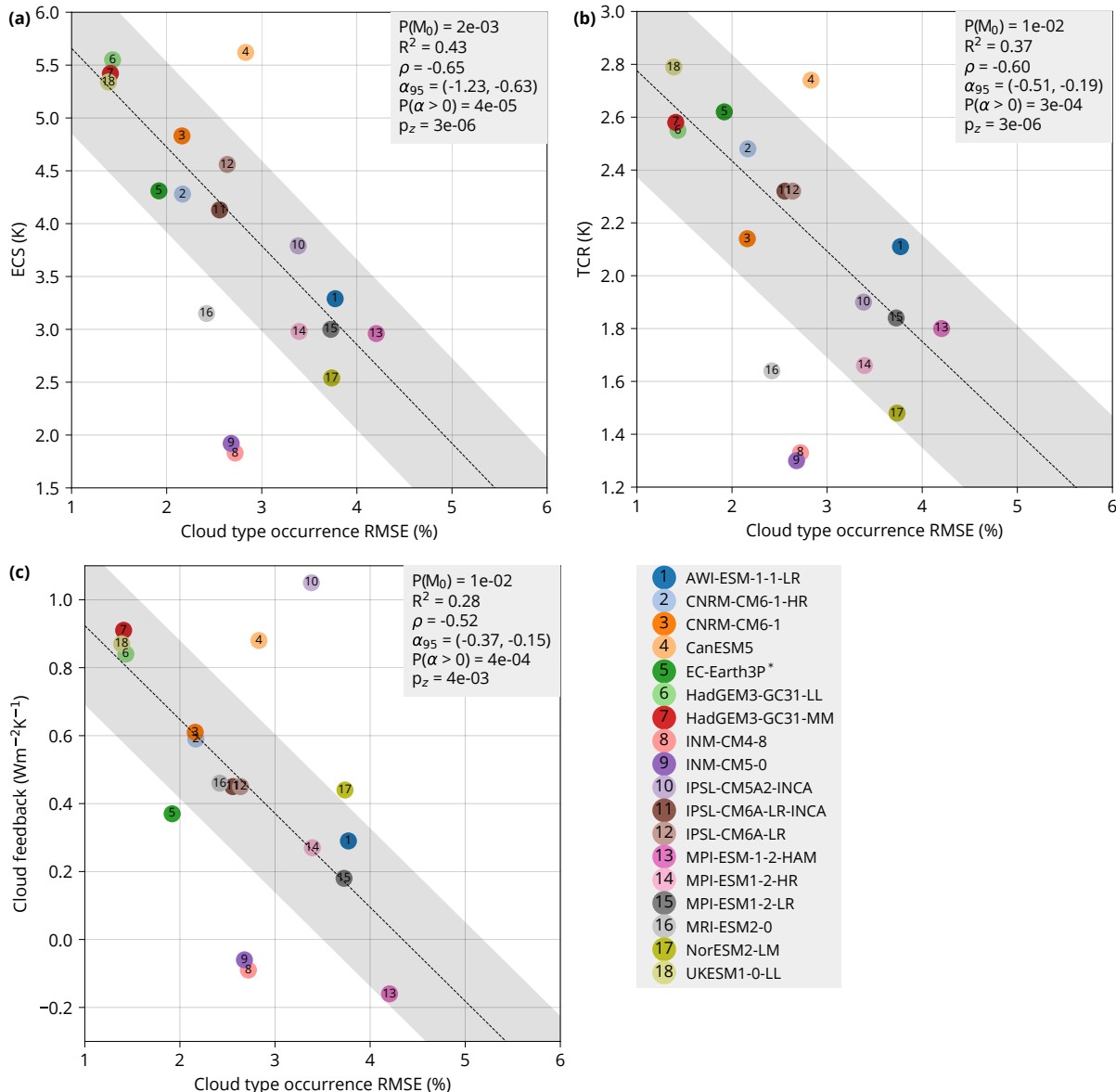

**Figure 12. (a)** Dependence of equilibrium climate sensitivity (ECS), **(b)** transient climate response (TCR) and **(c)** climate feedback of CMIP6 models on the model total cloud type root mean square error (RMSE) relative to CERES, calculated from the geographical distribution (as in Fig. 6 and 7). The points are are calculated from the ANN for 27 cloud types. Confidence bands represent 68% range. Linear regression is calculated using Bayesian simulation assuming Cauchy error distribution (Appendix A). Shown is also the probability of the null hypothesis model ($P(M_0)$) (explained in Sect. 4.6), the coefficient of determination ($R^2$), correlation coefficient ($\rho$), 95% confidence interval of the slope ($\alpha_{95}$) of the linear regression, probability that the slope is greater than zero ($P(\alpha > 0)$), and the p-value of a z-test ($p_z$) for the difference of means of two groups of models in the bottom 50% and top 50% of ECS (a), TCR (b) and cloud feedback (c). *For some models, ECS was not available and was taken from a related available model (see Table 1).

the $P(M_0) = 0.29, 0.29, 0.62$ for ECS, TCR and cloud feedback, respectively (Fig. S9), i.e. the presence of a negative linear relationship is still more likely than not for ECS and TCR, but not cloud feedback, although such a test is relatively weak due to the small number of remaining models.

The above was calculated with the ANN for the full set of 27 cloud types due to its higher statistical strength (as can be expected with a more detailed classification). If done with four and 10 cloud types, $P(M_0) = 4 \times 10^{-3}$, 0.01 and 0.07 (ECS, TCR and cloud feedback, respectively) for four cloud types (Fig. S10) and $P(M_0) = 3 \times 10^{-3}$, 0.01 and 0.06 for 10 cloud types (Fig. S11). This means that the relationship holds well even for the classifications with fewer cloud types, but with lower statistical strength.

## 5  Discussion and conclusions

We developed a deep convolutional ANN for the purpose of determining cloud types in retrieved and simulated TOA shortwave and longwave radiation images, trained on global historical records of human observations of WMO cloud genera (IDD). We trained this ANN to identify the probability of occurrence of each cloud type in every pixel of the image for a set of 4, 10 and 27 cloud types. We applied the ANN to satellite observations from CERES, CMIP climate model and reanalysis output to derive geographical distribution, global mean of cloud type occurrence and its change with respect to GMST. This provided a unique quantification of the distribution of WMO cloud genera globally, and enabled us to compare models and observations with this metric. Relative to IDD, the ANN could reproduce the geographical distribution of cloud type occurrence relatively well over regions where reference data were available. CMIP models and reanalyses displayed a variety of biases relative to satellite observations, most notably negative bias in the cumuliform cloud type and smaller negative bias in the middle and high cloud type. Models related in their code base often showed the same pattern and magnitude of biases. Models with lower ECS ($< 4$ K) had larger biases than models with higher ECS ($> 4$ K) and reanalyses. Analysis of the abrupt-4xCO2 experiment suggests that low ECS ($< 4$ K) models tend to simulate decreasing cumuliform and increasing stratiform clouds, while the opposite is true for high ECS ($> 4$ K) models. By linking the cloud type change with respect to GMST to ECS, we showed that models with decreasing stratiform and middle cloud type and increasing cumuliform cloud type tended to have higher ECS, a physically expected result. We investigated the link between present-day cloud biases and ECS, TCR and cloud feedback. We found that the model cloud biases are correlated with all three quantities. Models with smaller biases had higher ECS, TCR and cloud feedback than models with larger biases.

The method introduced in this study has a number of limitations. The CERES dataset is too short (2003 to present) to reliably detect change with respect to GMST. This means that we could not perform this kind of evaluation. It would be theoretically possible to perform such an evaluation with the CMIP historical experiment, which also includes the effect of aerosol, if a suitable satellite dataset were available. Because the ANN was not trained to be applied on pixels without SW radiation, polar regions during polar winter were not analysed. The analysed model ensemble was relatively small with several models of the same origin. Therefore, even though relatively strong statistical correlations could be identified, they rest on the assumption of statistical independence. In Fig. S9 we confirm that some of the identified associations hold on a smaller set of

unrelated models. Similar limitations as in past emergent constrains analyses apply, in that a physical explanation would need to accompany a statistical relationship for it to be confirmed.

The NOAA and ESA satellite series provide much longer time series than CERES. Alternatively, the ANN could be trained on radiation measurements from passive or active instruments other than normalised CRE from CERES, as long as they provide information about clouds which can be paired with ground-based station observations of cloud genera. Currently, the datasets derived from these satellite series, the Climate Change Initiative Cloud project (Cloud_cci) (Stengel et al., 2020), the Pathfinder Atmospheres Extended (PATMOS-x) (Foster and Heidinger, 2013) and the Climate Monitoring Satellite Application Facility (CM SAF) Cloud, Albedo And Surface Radiation dataset from the Advanced Very High Resolution Radiometer (AVHRR) data (CLARA-A2) (Karlsson et al., 2017), appear to be unreliable for determining change of clouds with respect to GMST due to changing orbit and instrument sensors. It is possible that future improvements will overcome these issues. Other satellite products which could provide suitable radiation information include ISCCP, Multi-angle Imaging SpectroRadiometer (MISR), MODIS, CloudSat or the Cloud-Aerosol Lidar and Infrared Pathfinder Satellite Observation (CALIPSO). The benefit of using other satellite dataset could be the confirmation of the results with an independent dataset, longer available time series, or the fact that the active instruments provide information about the vertical structure of clouds. This is a qualitatively different view of clouds than from passive instruments, which have only limited ability to detect overlapping clouds and determine the structure of thick clouds. For a direct comparison with models, an equivalent physical quantity is needed. A satellite simulator such as the Cloud Feedback Model Intercomparison Project (CFMIP) Observation Simulator Package (COSP) could be used to calculate such an equivalent quantity. In the case of normalised CRE, a simulator is not needed because it is a standard model output quantity.

When compared to the results produced by past clustering approaches based on SOM applied on ISCCP and MODIS (Appendix B), the ANN shows good agreement on the physical properties of clouds, but differences exist potentially due to multi-level cloud situations and low effective spatial resolution of the CERES/ANN dataset (about $5 \times 5°$). The definition of the ANN cloud types is also fundamentally different from previous methods because it is based on visual observations by humans, whereas other methods use a more synthetic approach of partitioning the cloud top pressure–optical depth space either directly (Rossow and Schiffer, 1991) or by machine learning methods such as SOM (McDonald and Parsons, 2018; Schuddeboom et al., 2018). The viewpoint also likely matters. Our method uses a hybrid top–bottom approach, where radiation fields measured from the top by satellites (or simulated by models) are used to derive cloud types corresponding to observations from the ground. Cloud top pressure–optical depth partitioning methods usually rely on radiation fields measured from the top only. Due to obscuration in multi-layer and thick cloud situations, top and bottom approaches can have very different views of reality (McErlich et al., 2021).

An important finding of our analysis is that cloud type occurrence biases in CMIP6 models show that more climate sensitive models are more consistent with observations in this metric. Zelinka et al. (2022) also recently found that the mean-state radiatively-relevant cloud properties in CMIP5 and CMIP6 models are correlated with total cloud feedback, and in particular that better simulating present-day cloud properties is associated with larger cloud feedbacks. They concluded that the explanation for this association is an open question for future research. In contrast with Zelinka et al. (2022) and our results,

Schuddeboom and McDonald (2021) did not find any relation between mean or compensating cloud errors and ECS in a cloud clustering analysis, although their model ensemble was small (8 models). The reason why their result is different from ours might be due to a number of factors, such as a small number of models analysed by Schuddeboom and McDonald (2021), a different set of models, their focus on SW CRE errors vs. our focus on the RMSE of cloud type occurrence probability, and a very different cloud classification method.

We suggest that our results showing that models with relatively high ECS perform better in the cloud type representation should be considered with caution. Limiting factors of our analysis were novelty of the method, limited validation options (Sect. 4.1) and the small size of the model ensemble. In addition, a credible physical mechanism needs to be established in order to confirm a statistical association. However, the result could be considered in the context of other factors influencing ECS in a multiple-factor analysis (Bretherton and Caldwell, 2020; Sherwood et al., 2020), especially if it should be used as an emergent constraint. Even though our results favour models on the high-end of ECS in the investigated model ensemble, they are not necessarily in contradiction with Sherwood et al. (2020) or AR6. Some of the models which performed well in our analysis lie on the upper end of the very likely range estimated in these reviews. The scope of our study is much smaller than either of the reviews, and utilises only one cloud metric on a limited number of related models. Nevertheless, we think that the strength of the identified relationship and the opposing trends in cumuliform and stratiform clouds in high ($> 4$ K) and low ($> 4$ K) ECS models with increasing GMST warrant a further investigation of links between present-day cloud simulation biases and projected future cloud change, and demonstrates that the ANN method of cloud identification can be a useful tool for climate model evaluation.

## Appendix A: Linear regression Bayesian model comparison

The linear regression model $M_1$ representing the alternative hypothesis and the null hypothesis model $M_0$ are defined as:

$$M_1: \quad y = \alpha x + \beta + \epsilon, \tag{A1}$$
$$M_0: \quad y = \beta + \epsilon, \tag{A2}$$
$$\alpha = \tan(\varphi), \tag{A3}$$
$$\varphi \sim \mathrm{Uniform}(-\pi/2, \pi/2), \tag{A4}$$
$$\beta \sim \mathrm{Uniform}(-100, 100), \tag{A5}$$
$$\epsilon \sim \mathrm{Cauchy}(0, \gamma), \tag{A6}$$

where $x$ is a vector of the independent variables, $y$ is a vector of the dependent variables, $\alpha$ and $\beta$ are the slope and intercept, respectively, $\epsilon$ is a Cauchy-distributed random error, $\gamma$ is the scale parameter of the Cauchy distribution, and $\varphi$ is the angle of the slope. $\varphi$ and $\beta$ come from a continuous uniform prior distribution. The statistical distribution of the free parameters $\varphi$, $\beta$ and the Bayes factor ($P(M_1|x,y)/P(M_0|x,y)$) were determined using the Metropolis algorithm (Metropolis et al., 1953), and simulated with the Python library PyMC3 version 3.11.2 (Salvatier et al., 2016). The prior probability of $M_0$ and $M_1$ was

590 assumed to be equal: $P(M_0) = P(M_1) = 0.5$. Before running the simulation, the variables $x$ and $y$ were normalised by their mean and standard deviation. For statistical significance we assumed $P(M_0)$ below 0.05.

## Appendix B: Comparison with cloud clusters derived using self-organising maps

To understand how the cloud types determined by the ANN relate to cloud clusters constructed by previous studies, we perform a comparison with cloud clusters of Schuddeboom et al. (2018) and McDonald and Parsons (2018), generated by a machine

learning method known as self-organising maps (SOM). They use SOM to identify representative cloud clusters using cloud top pressure–cloud optical depth joint histograms from Moderate Resolution Imaging Spectroradiometer (MODIS) (Schuddeboom et al., 2018) and ISCCP (McDonald and Parsons, 2018). They establish characteristics of these clusters by investigating how they relate to cloud properties. Schuddeboom et al. (2018) uses the clusters to examine model representation of different cloud types while McDonald and Parsons (2018) focuses specifically on how their clusters relate to atmospheric dynamics.

Here we calculate the average CERES/ANN values for each of our four types for every ISCCP/SOM and MODIS/SOM cluster. The CERES/ANN geographical distribution is available on a $5°\times5°$ global grid (the original grid is $2.5°\times2.5°$, but the effective resolution is lower), while the ISCCP/SOM data are on a $2.5°\times2.5°$ grid and MODIS/SOM data are on a $1°\times1°$ grid. To account for this difference in spatial resolution, all of the ISCCP/SOM and MODIS/SOM grid cells that fall within a corresponding ANN geographical distribution grid cell are considered as having the same occurrence values. This will

overestimate the similarity between the clusters, as the small cloud structures that can be identified in the higher resolution dataset will be merged.

In Fig. B1 we present calculated co-occurrence of the CERES/ANN cloud types with the 15 ISCCP/SOM clusters of McDonald and Parsons (2018) and 12 MODIS/SOM clusters of Schuddeboom et al. (2018). The left–right and top–bottom ordering of the ISCCP/SOM and MODIS/SOM grids is the result of the SOM algorithm, from which these clusters were derived. This

algorithm results in neighbouring clusters which are closely related, with the most distinct clusters the most distant. For example, in the ISCCP/SOM grid the top row relates to clouds with low cloud top pressure while the bottom row relates to high cloud top pressure. From understanding this relationship, we can see that ordering of the CERES/ANN values suggests good separation into physically distinct cloud types. The values shown suggest small to moderate amounts of every cloud type are present regardless of the ISCCP/MODIS cluster present. This could be at least partially explained by the spatial smoothing

effect described above, as well as co-occurrence of different cloud types in a single geographical grid cell.

By considering individual clusters in Fig. B1 and examining their cloud top pressure–cloud optical depth diagrams (McDonald and Parsons, 2018, Fig. 1) and (Schuddeboom et al., 2018, Fig. 2), we can see that they show the expected physical relationship. The CERES/ANN high cloud type is identified as co-occurring most strongly with ISCCP/SOM clusters 1, 6, 7, and 11, which are also the SOM clusters corresponding to high clouds (McDonald and Parsons, 2018, Fig. 1). The CERES/ANN

middle cloud type is most strongly associated with ISCCP/SOM clusters 2, 1 and 4 (numbers ordered by the strength of association), which all contain substantial amount of semi-opaque clouds at 180–680 hPa. The CERES/ANN cumuliform cloud type is most strongly associated with ISCCP/SOM clusters 13, 6, 11 and 12. While cluster ISCCP/SOM 13 has a local maximum at

low cloud of low-to-mid optical depth, clusters 6, 11 and 12 are less clearly associated with low clouds (they have maximum for high low optical depth clouds), although they still contain substantial amounts of low altitude low optical depth clouds. The CERES/ANN stratiform cloud type is most strongly associated with the ISCCP/SOM clusters 4, 5, 2, 10 and 15. The ISCCP/SOM clusters 10 and 15 are strongly associated with low-altitude mid-to-high optical depth clouds, as expected for stratiform clouds. The ISCCP/SOM clusters 4 and 5 are mostly composed of mid-altitude mid-optical depth clouds. The ISCCP/SOM cluster 2, however, is mostly associated with relatively high clouds above 680 hPa.

The CERES/ANN high cloud type co-occurs most strongly with the MODIS/SOM cluster 1. This cluster has maximum for high-altitude low optical depth clouds (Schuddeboom et al., 2018, Fig. 2) and is identified as tropical. The CERES/ANN middle cloud type is most strongly associated with the MODIS/SOM clusters 9 and 10, identified as mixed level clouds. The MODIS/SOM cluster 9 has the greatest contribution from relatively high clouds above 310 hPa, but substantial amount of clouds at altitudes of 180–800 hPa. The MODIS/SOM cluster 10 has greatest contribution from clouds at relatively low altitude of 680–800 hPa, but also high clouds above 440 hPa. The CERES/ANN cumuliform cloud type is associated with the MODIS/SOM clusters 3, 1, 2, 4, 7 and 8. These are identified as marine or tropical, and have strong contribution from low-altitude low-to-mid optical depth clouds. The CERES/ANN stratiform cloud type is associated with MODIS/SOM clusters 10, 12, 9 and 11. These are identified as mixed level and stratocumulus clouds, and have a strong contribution from low altitude mid-to-high optical depth clouds (clusters 11 and 12) and clouds at various altitudes (clusters 9 and 10).

To summarise, the correspondence between the CERES/ANN cloud types and ISCCP/SOM and MODIS/SOM clusters is relatively good when compared using the cloud top pressure–cloud optical depth diagrams. However, differences exist particularly in cloud types related to mixed level cloud situations.

*Code and data availability.* The datasets used in our analysis are publicly available: CERES (2022), GISTEMPv4 (GISTEMP Team, 2021), CMIP5 (2022), CMIP6 (2022), MERRA-2 (2022), ERA5 (2022) and IDD (Unidata, 2003). The code used in our analysis is open source and available on GitHub (Kuma, 2022) and Zenodo (Kuma et al., 2022).

*Author contributions.* PK participated on conceptualisation and methodology development, developed the artificial neural network model, performed the data analysis and wrote the manuscript; FB participated on conceptualisation and methodology development, review and editing of the manuscript, funding acquisition and project administration. AS and AM performed the comparison with MODIS and ISCCP cloud regimes and reviewed the manuscript. ØS participated on NorESM2-LM data preparation and acquisition, consultation of the analysis and reviewed the manuscript.

*Competing interests.* The authors declare that they have no conflict of interest.

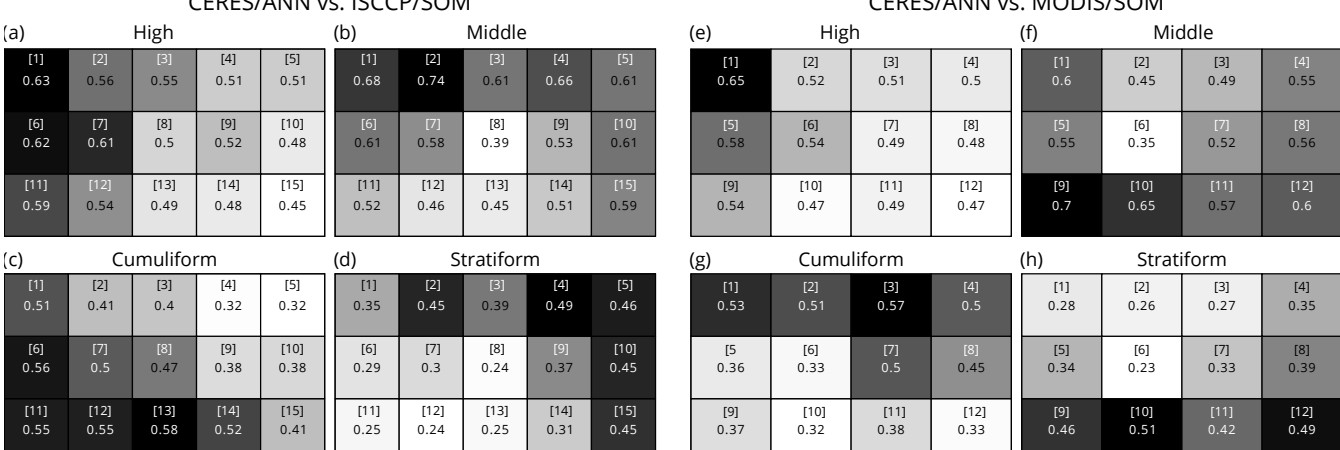

**Figure B1.** Comparison of co-occurrence of our ANN-derived cloud types (CERES/ANN) with SOM-derived cloud clusters (ISCCP/SOM and MODIS/SOM) of McDonald and Parsons (2018) and Schuddeboom et al. (2018), respectively. The grids in **(a–d)** and **(e–h)** represent our four ANN-derived cloud types (high, middle, cumuliform and stratiform). The grid boxes in each subplot correspond to the SOM-derived cloud clusters of the past studies. Numbers in square brackets in the boxes are the ISCCP/SOM and MODIS/SOM cluster numbers as in the original studies. Numbers in the centre of the boxes and box shading are the co-occurrence (scale 0–1) of the ANN-derived cloud type and the SOM-derived cloud cluster. The co-occurrence is calculated from one year (2007) of daily mean values on a global spatial grid. Note that the definition of the ISCCP/SOM and MODIS/SOM clusters is different, and therefore **(a–d)** and **(e–h)** are not expected to be similar.

*Acknowledgements.* This work was conducted as part of the FORCeS project: 'Constrained aerosol forcing for improved climate projections' (FORCeS, 2022), funded by the European Union's Horizon 2020 research and innovation programme under grant agreement number 821205. We also acknowledge funding from the Swedish e-Science Research Centre (SeRC). AS and AM acknowledge funding from New Zealand's Deep South National Science Challenge 'Cloud and Aerosol Measurements for Improved Climate Projections'. We thank Hossein Azizpour
for his consultation and advice on the artificial neural network development. We acknowledge the CERES dataset provided by the NASA Langley Research Center, the IDD dataset provided by Unidata and the University Corporation for Atmospheric Research through the Research Data Archive at the National Center for Atmospheric Research, the GISTEMP dataset provided by the NASA Goddard Institute for Space Studies, the ERA5 dataset provided by the ECMWF through the Copernicus Climate Change Service, and the MERRA-2 dataset provided by the Global Modeling and Assimilation Office, NASA Goddard Space Flight Center Greenbelt. We acknowledge the World
Climate Research Programme, which, through its Working Group on Coupled Modelling, coordinated and promoted CMIP5 and CMIP6. We thank the climate modelling groups for producing and making available their model output, the Earth System Grid Federation (ESGF) for archiving the data and providing access, and the multiple funding agencies who support CMIP5, CMIP6 and ESGF. We acknowledge open source software used in our analysis: TensorFlow (Abadi et al., 2016), Python, NumPy (Harris et al., 2020), SciPy (Virtanen et al., 2020), Matplotlib, cartopy (Met Office, 2010), PyMC3 (Salvatier et al., 2016), parallel (Tange et al., 2011), Pandas (The pandas development team,
2020), pyproj, Cython (Behnel et al., 2011), aria2 and Devuan GNU+Linux.

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
