# Peer review of "Machine learning of cloud types in satellite observations and climate models"

_Atmospheric Chemistry and Physics, 2022_

## Referee Comment (RC2)

Review of "Machine learning of cloud types shows higher climate sensitivity is associated with lower cloud biases" by Kuma et al MS No.: acp-2022-184

**Summary**

The authors train a supervised deep convolutional artificial neural network to predict the frequency of occurrence of four human-observed cloud types based on the CERES-measured top of atmosphere longwave and shortwave radiation fields over a large 4000 km x 4000 km region. After validating its ability to reproduce the observed cloud types on data withheld during training and comparing their results with an independent cloud categorization analysis, they apply the algorithm to climate model output, thereby allowing them to evaluate the fidelity with which models simulate the various cloud type occurrence frequencies. Model skill in simulating these current-climate cloud occurrences is assessed in light of the model climate sensitivities (ECS and TCR) and cloud feedbacks, and it is found that more sensitive models tend to have smaller mean-state cloud errors, although the cloud feedback shows little relationship with mean-state cloud errors. The authors argue that the most likely explanation for their results is that high ECS is plausible, in contrast to recent expert assessments (Sherwood et al. 2020; Masson-Delmotte et al. 2021)

I find the paper's overall goal to be interesting and worthwhile, but I had substantial difficultly following it in several sections and I believe the authors need to be a little more circumspect with the interpretation of the results. I recommend major revisions, as detailed below.

**Comments**

- 1. I had a very hard time following what was done in setting up, training, and validating the ANN. This includes both understanding it at the conceptual level and in many of the details. I think the authors need to begin Section 2.2 with a "30,000 foot" view of what they are trying to do, namely, predict the frequency of occurrence of 4 WMO cloud types within a 4000 km x 4000 km box based on the (spatial pattern of?) TOA radiative fluxes observed within that box. I think some interpretation of what information the ANN is learning from is needed. Is it the spatial pattern / orientation of SW and LW radiation within the region, the regional-average values, or something else entirely that provides the needed information? Are the SW and LW information equally important or does one band provide most of the information? Why is a deep convolutional artificial neural network needed in the first place; what is it providing that simpler methods would fail to yield? Table 2 and Algorithm 1 are utterly incomprehensible to me, and probably to a majority of readers of this journal. These details should probably go in a supplementary materials or an appendix, and they should be replaced with something more like schematics that give a sense of the basic workflow, how the ANN is set-up, how the data is split among training and testing, etc. All of these details are hard to extract from the paper.
- 2. Is it wise to use random selection for splitting the training and testing subsets? I would assume that there could be some autocorrelation in the data that would cause such an approach to overstate the skill relative to a situation in which, say, the (chronologically) first 80% of the data is used as training and the last 20% of the data is used for testing, etc.
- 3. What does it mean that the ANN explains 47% of the variance? Variance in what? And where does this number come from; is it in a figure? On line 407, it is stated that this number is "relative to an uninformative predictor" but elsewhere it is not stated relative to anything. How do I interpret this?
- 4. Figures 4 and 5: "Stratiform" is misspelled

- 5. Section 3.3.: How did you calculate these joint histograms? I can't find any details on how this is done or what data is used in the methods section.
- 6. Section 3.4: I am completely lost in this section on comparing to MODIS and ISCCP cloud clusters. How is the ANN now being generated on a 5x5 degree grid when previously the highest resolution is 4000 km x 4000 km? What exactly are you showing in Figure 7? I assume the reader has to be familiar with Schuddeboom et al. (2018) and McDonald and Parsons (2018) to understand this, but how many readers will be? I read this several times and I simply cannot wrap my head around what is being done here, other than a vague sanity check that what the ANN calls "high", "stratiform", etc. is consistent with independent cloud clustering methods. I recommend a complete re-write of this section keeping in mind that the average reader is not familiar with these other studies.
- 7. Section 3.5: I do not see any justification for regressing the observed cloud types on global mean surface temperature during the brief CERES record that (1) it is likely dominated by internal variability that is not directly relevant to the long-term cloud feedback and (2) likely includes effects related to changes in aerosols and other non-CO2 forcings. Placing these results side-by-side with the abrupt-4xCO2 cloud changes is misleading and not a robust evaluation of models. You already note that this is not a "reliable observational reference", so I wonder why you did it. A more minor point: the abrupt-4xCO2 simulation does not occur during a particular time in the historical record (noted here as 1850-1949) but rather to an arbitrary 150-year period whenever the modeling center decided to branch from its piControl simulation. So I think you meant to say simply that you used data from the 150-year experiment.
- 8. Bayes factors: Maybe I am just ignorant, but this is the first time I had ever seen these numbers for significance testing. Perhaps other readers will also be clueless about what these numbers mean. Please provide some brief explanation when these first appear in the text, and discuss in more detail their meaning in the appendix.
- **9.** Lines 321-323: The choice to report the ECS values for RMSE values below an arbitrary threshold (2.4%) is a little egregious, given that if the threshold for what is considered "low" RMSE is relaxed only slightly (to say 5%), the entire range of ECS values is now supported. Ditto for the "high" RMSE values: If you take all models with higher-than-average or even probably the models in the top 10 percentile of RMSE, you will include the high-ECS CanESM5 model.
- 10. Figure 10b: I wonder how large the Bayes factor would be if models from the same modeling center were averaged together before computing significance. Would the relationships derived in the paper between cloud occurrence RMSE and ECS remain so strong once the 3 UKMO models, 2 CNRM models, 2 INMCM models, 3 IPSL models, and 3 MPI models are combined? This would represent a substantial decrease in sample size from 18 to 10.
- 11. Lines 337-340: I find this reasoning for why simulating good mean-state clouds should translate to simulating good clouds in a future warmed state to be dubious. It is likely that clouds will inhabit an environment with different conditions in the future (e.g., one with higher SSTs, stronger inversions, and a sharper moisture contrast between the boundary layer and free-troposphere) refer to the cloud controlling factor literature (Bretherton 2015; Klein et al. 2017).
- **12.** Line 347: "lower ECS" is a little misleading, as I think you mean lower than the highest ECS values in CMIP models, but not lower than, say, the canonical IPCC range.
- **13.** Lines 353-354: In my opinion the simplest / most likely explanation is neither of these, but rather that you are looking at a very small sample size of models (especially once you combine closely related models from the same center) and spurious correlations can occur. Perhaps more importantly, I am led to doubt the robustness of the correlation with ECS because the correlation with cloud feedback is poor (Figure 10d). How are we to believe that accurately simulating mean-state clouds translates to a better representation of ECS if the most obvious intermediary (cloud feedback) shows no relationship with mean-state cloud quality?

- **14.** Lines 400-404: I don't understand what is being suggested here or how it could be used in concert with the techniques employed in this study.
- **15.** Lines 432-435: My read of Zelinka et al (2022) is that the quality of present-day cloud representation has very little bearing on the quality of its cloud feedback (see their Figure 4b). Seems worth mentioning this, rather than the weaker statement that it is an open question.
- 16. Lines 435-440: Somewhere around here it may bear mentioning the notion that emergent constraints based on mean-state climatological observables (like the occurrence frequency of the 4 cloud types in this study) are generally less useful or robust than those that narrow in on processes relevant to the climate change phenomenon of interest (Klein and Hall 2015; Hall et al. 2019)

**References**

- Bretherton, C. S., 2015: Insights into low-latitude cloud feedbacks from high-resolution models. *Philosophical Transactions of the Royal Society a-Mathematical Physical and Engineering Sciences*, **373**, https://doi.org/10.1098/rsta.2014.0415.
- Hall, A., P. Cox, C. Huntingford, and S. Klein, 2019: Progressing emergent constraints on future climate change. *Nature Climate Change*, **9**, 269–278, https://doi.org/10.1038/s41558-019-0436-6.
- Klein, S. A., and A. Hall, 2015: Emergent Constraints for Cloud Feedbacks. *Current Climate Change Reports*, **1**, 276–287, https://doi.org/10.1007/s40641-015-0027-1.

---, ---, J. R. Norris, and R. Pincus, 2017: Low-Cloud Feedbacks from Cloud-Controlling Factors: A Review. *Surveys in Geophysics*, https://doi.org/10.1007/s10712-017-9433-3.

- Masson-Delmotte, V., and Coauthors, eds., 2021: Climate Change 2021: The Physical Science Basis. Contribution of Working Group I to the Sixth Assessment Report of the Intergovernmental Panel on Climate Change. Cambridge University Press,.
- Sherwood, S. C., and Coauthors, 2020: An Assessment of Earth's Climate Sensitivity Using Multiple Lines of Evidence. *Reviews of Geophysics*, **58**, e2019RG000678, https://doi.org/10.1029/2019RG000678.

---

## Author Comment (AC1)

**Response to Referee 1 on manuscript 'Machine learning of cloud types shows higher climate sensitivity is associated with lower cloud biases'**

Peter Kuma, Frida A.-M. Bender, Alex Schuddeboom, Adrian J. McDonald, and Øyvind Seland

July 22, 2022

Dear Dr. Steven Sherwood,

Thank you for your insightful comments. Please find below our response. In the following text, the original comments are in **bold**, followed by our response. We do not provide a document marking differences between the original and revised manuscripts, because the changes were too substantial for it to make sense.

In the revised manuscript we use a new version of the ANN which performs pixel-wise classification. We hope that this change will address some of the main concerns. We also present a classification for 10 and 27 cloud types, which provides more nuanced results than what would be possible with more traditional cloud classifications. We changed the manuscript title to 'Machine learning of cloud types in satellite observations and climate models' to emphasise the new method of cloud classification over of the implications on climate sensitivity.

Kind regards,

Dr. Peter Kuma on behalf of the authors

**General Comments**

**This paper uses an artificial neural network (ANN) to learn cloud populations (via four cloud types) from top-of-atmosphere LW and SW cloud-radiative effect, demonstrating moderate skill. It then uses this as a metric for comparing climate model cloud fields to observations, by computing RMS error between the satellite-derived and model-derived distributions of the cloud types. This is in principle a very interesting idea, because it provides an arguably more objective way of identifying observational metrics. The authors report a decreasing trend in RMS error with increasing climate sensitivity, which they argue implies that ECS is high in spite of recent assessments finding this to be very unlikely.**

In the original manuscript, we present alternative explanations for the relationship between the error and ECS. The more obvious explanation of this is that models with better present-day simulation of clouds are also better with their representation of cloud feedback and therefore ECS, but we do not present this as the only possible explanation. Only the models UKESM/HadGEM and CanESM5 have ECS which is very unlikely in AR6 (above 5 K). Other models which still performed relatively well in our analysis have ECS below 5 K and are still in the very likely range of AR6.

**I believe this work requires major revisions before it should be published, and I'm not sure it should be published at all unless some of these concerns can be satisfactorily overcome (or I have something wrong that just needs explaining better), which hopefully they can. The main problems are elaborated below.**

**Specific Comments**

**The conclusions drawn are not convincing given the limited number of independent models and the modest strength of relationships seen in Fig. 10. For example if you took out the three MPI models, or the three UKMO models, in either case the correlation would become pretty weak. The authors are using a Cauchy error model which is more forgiving of outliers (of which there are several) than the more common Gaussian model, and as they do not specify otherwise I assume they are taking the models to be independent (which is not a good assumption since, as the authors**

themselves discuss, various models by the same centre often behave similarly). **This problem could be addressed if the authors were able to include more models (and I am surprised more cannot be used).**

With the revised ANN the results are strengthened. In the revised manuscript we skip some of the discussion related to ECS and point out the limitation related to the size of the model ensemble more clearly. We perform a sensitivity test in which we remove all closely related models. The results of the test still point to the linear relationship more likely than not for ECS and TCR.

Cauchy distribution is not only more forgiving to outliers, but it is also harder to statistically 'prove' a linear relationship vs. no relationship. We think that it is general a better distribution to assume than normal distribution for model ensembles. In model ensembles, the assumption of normally distributed values is unlikely to be appropriate. This is because often configuration or code issues can result in outliers, i.e. there is a relatively small number of factors which can cause large effects and the central limit theorem is unlikely to apply well.

Other model ensemble studies and reviews also often assume model independence, and therefore share this limitation. As stated in the original manuscript, we included all CMIP5 and CMIP6 models which provided the necessary variables and no subjective selection was done.

**Certainly, given that several models disobey the fit, this leaves a reasonable chance that the real world would also disobey the fit, whereas the authors seem to be tacitly assuming in their discussion that the only way ECS could be anything but very high is for the whole relationship to be perversely wrong.**

We disagree with this characterisation of our original manuscript. We do say that based on our results, the more obvious interpretation is that high ECS (high in the context of our model ensemble) is more correct. But we clearly mention the other interpretation and discuss limitations of the evaluation.

In the revised manuscript, fewer models are outliers. The revised ANN produces results which are more clearly related with ECS, TCR and cloud feedback. We also fixed data processing issues with the NorESM2 and CanESM5 models. The only outliers are now only INM, CanESM5 and MRI-ESM2 for ECS and TCR, and also IPSL-CM5A2-INCA for the cloud feedback. It is perhaps likely that the value for cloud feedback of IPSL-CM5A2-INCA (from an external source) is erroneous since it is very far from the other IPSL models (and the highest of all models in the ensemble).

**Having said this I do think it is useful and interesting to show that the more accurate models have higher ECS (although that has been shown by other studies using different metrics), since even if this doens't prove ECS is high, it does identify a conundrum that the modeling community needs to solve.**

**Also the authors should be aware of and probably cite papers such as Zhu et al. 2022 doi:10.1029/2021MS002776, who found that the NCAR model could be improved by making a change to the cloud scheme which also reduced the ECS of the model, i.e., a counterexample to the claimed relationship, or Zhao et al. 2016 (10.1175/JCLI-D-15-0191.1) who found that the ECS of the GFDL model could be changed substantially without affecting the latitudinal distribution of cloud radiative effect.**

In the discussion and conclusions, we added: 'Zhao et al. (2016) showed that it is possible to modify parametrisation of precipitation in convective plumes in the GFDL model and get different Cess sensitivities without increasing CRE error relative to CERES. Zhu et al. (2022) showed that in CESM2, a CMIP6 model with very high ECS of 6.1 K, a physically more consistent cloud microphysics parametrisation reduced the ECS to about 4 K and produced results more consistent with the last glacial maximum.'

**Other authors (Klein, Hall, Caldwell) have pointed out that emergent constraints should be treated with much caution unless there is a mechanism linking the observable to the feedback; simply pointing out that clouds are being observed and are involved in the feedback is not a mechanism.**

Even though we do not use the metric as an emergent constraint, we recognise that it also has this limitation. We point out this limitation in the revised discussion and conclusions.

**Finally, while the authors are entitled to their opinion on how much credibility to give their analysis vs. that of the IPCC and Sherwood et al. (which they should not call an "expert judgment" study since that implies it was based on an expert elicitation rather than an analysis of evidence), I would say it is unfairly dismissive given that those assessments quantitatively incorporated many independent lines of evidence including other emergent constraint studies not dissimilar to this one, for example Volodin et al. which is arguably based on a similar argument and dataset to the current paper (and still has some limited skill — see Schlund et al. 2020). The constraint on ECS offered by**

the authors is much more indirect and model-dependent than the multiple additional constraints used by the other assessments.

We do not think that our results invalidate AR6 or Sherwood et al. (2020) for several reasons. Only the HadGEM/UKESM and CanESM5 models in our analyses have ECS above the upper very likely bound of AR6 (5 K), and therefore can be considered inconsistent with AR6. We only use one type metric to evaluate the models, which cannot compete with much more extensive reviews. The method itself has various limitations (as mentioned in the original manuscript). There can also be decoupling between the accuracy of simulation of present-day and future clouds. The conclusions in the original manuscript were already worded carefully for this reason. Some of the conclusions were also based on results presented in the manuscript rather than all available evidence as done in AR6 or Sherwood et al. (2020), as this would be impossible to do fully in a relatively short manuscript whose focus is on introduction of a new evaluation method. The aim is to perform model evaluation which could potentially be considered alongside other evaluation methods in reviews like AR6 or Sherwood et al. (2020). In the revised manuscript, we skip much of the discussion related to low and high ECS models.

We call the review of Sherwood et al. (2020) 'expert judgment' because this is how Zelinka et al. (2022) referred to their analysis in the title of their paper and this term is also used in Sherwood et al. (2020). It was not meant to convey any kind of opinion. In the revised manuscript we do not use this term any more.

**I am not convinced that the ANN is behaving as expected. First, the authors have not shown any spatial maps of their verification data to compare with the maps of predicted cloud types. Second, the optical-depth/cloud-height histograms (Fig. 6) don't make any sense—they show that high clouds have the same distribution as the overall mean, but two different low-cloud types differ in opposite directions. But the high-cloud composite should show more, er, high cloud. I don't think this can be right.**

In the revised manuscript we present a comparison with maps derived from IDD (Fig. 3, supplementary plots and Section 3.4).

The cloud top pressure-cloud optical depth plots (Fig. 5) show more clear separation between the cloud types with the revised ANN. In Section 3.3, we contrast the results with a classification produced for ISCCP (Rossow and Schiffer, 1991).

**The cloud dataset is not adequately described. Are cloud amounts of each type given in oktas? Or are the clouds seen at each time simply assigned to one of the 27 categories? Or can multiple categories be assigned in a single synoptic observation, i.e., each category assigned a one or zero at each observing time? What exactly is the ANN going to predict? This needs to be given in Section 2.1.4.**

We added the description of IDD in Section 2.2.4. Cloud amounts are not taken into account, only yes/no observation of a cloud genus/species for the three heights (low, middle and high). This is encoded as three numbers denoting the cloud genus/species category (or missing) in every station report. The ANN predicts the probability of positive observation of a given cloud type.

**The way the ANN is employed is also not adequately explained, I'm not sure I fully understand what the authors have done, and what I do think I understand, I mostly had to piece together based on strands in the discussion of results. Also the motivation for the experiment design is not explained — what do we expect this approach to gain that was not gained by all the other efforts to classify or divide cloud scenes into different categories? You need a new section before "Results" where you explain the methodology properly. In 3.1 you don't say enough. For example you need to explain exactly what the features are and what you are trying to predict (see above comment, we don't even know the cloud states are represented via numbers). Are you (as I suspect) presenting each ~50x50 grid as single training instances? In which case the output of the ANN is a same-size grid of some measure of cloud state (Predicted cloud type, amount of each cloud type—don't even know if this is a categorical (classifier) or a real-valued (regression) target variable). Or are you prediction how many grid points will be assigned to each cloud type? Are any other variables used as training features, for example surface temperature (which you said earlier you were using but I don't see where it comes in)? Or the only predictors are the grid of normalised SW and LW CREs? As I understand it, according to Fig. 1, most of these grid points will have no verification value available, only those observed by an IDD station. I assume you then retain the predictions only at those locations? Help!**

In the revised manuscript we give an additional overview of the method in Section 2.1 and expanded Section 2.3 describing the ANN. Some of the concerns are addressed by pixel-wise classification in the revised ANN and the presentation of results for 10 and 27 cloud types, which go beyond what would be possible with more traditional methods.

**If I understand correctly, I don't think the authors are doing this in an optimal way. Based on my inferences, I believe each training instance is a tile of roughly 50x50 grid points; that the features are the gridded maps of SW and LW (normalised) CRE; and that the target variable predicted is the amount of each cloud type in the tile. This means the ANN predicts only four numbers for each tile, thus the authors can make only extremely smooth estimates (e.g. Fig. 4) with no detail at or below the tile dimension. Yet one should be able to predict high, mid and low cloud quite effectively on a ocal basis just from local SW and LW CRE: high LW CRE means high clouds, high SW and low LW CRE means low clouds; etc. Indeed this is routinely done and is the basic for e.g. the ISCCP cloud classification and other similar ones. It is not clear whether, or how this study has used any of the nominally available spatial information in the tile. If it is not being used then there is no point in starting with tiles, why not use each grid point (where you have a verification datum) as a training instance? If the authors do want to use spatial information (i.e. texture in the cloud field), then I would expect the authors to train on the entire global dataset using a convolutional neural network or other image processing approach that can produce detailed localised predictions rather than only producing populations accumulated over a large region. Or indeed they could use many more (and probably smaller) tiles in a standard NN and predict only the cloud properties at the centre of the tile. This would enable much more incisive testing of both the algorithm and the climate models.**

The revised ANN produces 4, 10 and 27 numbers (for a set of 4, 10 and 27 cloud types) for each 2.5° grid cell for samples of 16×16 pixels and 4000×4000 km. The geographical distribution (Fig. 3 and 4 and supplementary plots) now provides much more detailed information. Comparison with the ISCCP cloud classification is in Section 3.3.

**I don't find the comparison to traditional classification (Fig. 7) to be useful because of the way the data are being so severely coarse-grained (see point 4). The traditional measures are all local. The comparisons seem to suggest that their classifications have little to do with the traditional ones, which again is highly suspicious since the latter are based on robust physical arguments and should work well at least for high/med/low cloud distinction.**

This point should be addressed by the fact that the revised ANN produces much higher resolution output (2.5°, but effective resolution is about 5°). In the revised manuscript, with compare with the cloud top pressure-cloud optical depth classification of Rossow and Schiffer (1991) (Section 3.3) and keep the comparison with SOM-derived classifications (Appendix B). The comparison results show relatively good correspondence with both, but important differences exist.

**Technical suggestions**

**Section 2.1.2: I'm surprised that so few models are able to be included, since nearly all will have done the two required experiments, and yet you have fewer than half of the models. I assume that most models did not provide all of the desired radiation variables. Can you specify what was the main thing missing from the other models?**

Many models were missing daily mean outgoing clear sky shortwave or longwave radiation (rsutcs, rlutcs). We included all CMIP5 and CMIP6 models which supplied the required variables.

**Fig. 1: caption states that panels show spectra, but a spectrum is a graph of intensity vs. wavelength. These panels show images not spectra. Also, is the GCM image also from a single day? What day? how was it chosen? It looks fairly close to the observed one so I am guessing you searched somehow for a day that was close—if so this needs to be explained.**

By spectrum we mean the spectrum of the electromagnetic radiation. We replaced the term 'spectrum' with 'radiation' everywhere in the revised manuscript. All GCMs in our dataset are free running, and therefore do not simulate real weather patterns. The day is indicated at the top of the figure: '2010-01-01'. There was no particular reason for choosing this day. It was chosen simply as the first day of a year in the middle of the time period analysed. There was no search for a matching image involved because Fig. 1b and 1c and not to be compared directly – Fig. 1b is real while 1c is a free running model (in the revised manuscript this is Fig. 2).

**158: I don't understand what is being assumed multivariate normal here, or even what the three dimensions are (lon/lat/time?). Why do we assume a normal distribution in time? I would have thought we were just grabbing data from each day, at uniformly (not normally) sampled tile locations.**

The samples were uniformly sampled on a sphere. We used a normal distribution to derive the uniform sampling. One method of how to generate uniformly distributed samples on a sphere is to generate points from a multivariate normal distribution

in Cartesian coordinates, discard distance from the centre and only keep the angles. We skip this description in the revised manuscript because it is not very important beside the fact that that the samples were uniformly distributed.

**159: please also specify that these are TOA upward values**

We did so in the revised manuscript.

**170: please clarify if you discarded these points both for SW and LW training, or only for SW. I don't see any reason to discard them for LW, unless it would make your ML approach inefficient to have unpaired LW values. Clouds in the polar night will have nontrivial LW forcing effects.**

They were discarded if missing values occurred in either SW or LW. The reason for this is because the ANN does not accept missing values and inserting zeros as a replacement could affect the prediction. A workaround could be to train the ANN separately for only the LW channel and join the results, but we did not do this because of time constraints. It is true that this can affect the results at polar latitudes. We try to make this clear in the text.

**Fig. 3: Do I correctly interpret from this figure that the ANN is performing better on the evaluation partition than the training partition of the data? That is not usual. Maybe there is something I am missing here?**

The training and validation datasets have different time coverage and as a result also geographical coverage. In the revised ANN, the validation dataset also has lower loss function (Fig. S1). For a different choice of years for the training and validation datasets, the opposite is true. Some years are missing up to three weeks of data (those years which miss more were excluded), which means their coverage of stations is different. In the revised ANN, the loss function is calculated as the sum of log-likelihood from all pixels in all samples with one or more available stations reports.

**199: when you say "Cloud types in … reanalyses", I assume you mean the cloud types inferred by running the TOA radiation data through the ANN, not the actual clouds in the reanalysis? Please clarify.**

Yes, that is the case – cloud types inferred by running the TOA radiation data through the ANN.

**201-4: This is not a complete sentence (no verb)**

We reformulated this paragraph to reflect the new results.

**205: I don't think you can dismiss the error that easily. There are strat-cu decks that would not have any other cloud type, and would be large enough to be resolved. I do see them, smoothed out, but to tell if they are fully represented or not we'd need a comparison truth plot.**

The revised ANN can now resolve the start-cu decks. This is visible in Fig. 3 a4 and in the supplementary plots for 10 and 27 cloud types.

**Fig. 4: Why doesn't this figure include a set of panels for the target (IDD) observations? Only then can we see whether the algorithm is working, no? (see point #2 above)**

We added a row for IDD. The IDD data are only available where there are any stations. We keep only pixels with enough coverage of the whole year. Section 3.2 describes comparison between CERES/ANN and IDD.

**224: wrong citation format**

We corrected this in the revised manuscript.

**Fig. 7: we need to be told what the three rows and five columns represent. No good to just cite a paper we have to go look at to find out. More generally, I am not sure this whole analysis adds much to the paper anyway (see point #6 above).**

We added more description in Section 3.6, so that the analysis can be understood without reading the original papers. We moved the section to an appendix (Appendix B) to de-emphasise it and removed Fig. 8.

**Fig. 8: The caption doesn't fully clarify what is different between the second and third rows. Is the third row the histogram of 5x5 degree averages? The others are histograms at the original scale (1 degree?)**

We removed Fig. 8 because it now does not add much information with pixel-wise classification, which has the same spatial resolution as in the input data.

**Response to Referee 2 on manuscript 'Machine learning of cloud types shows higher climate sensitivity is associated with lower cloud biases'**

Peter Kuma, Frida A.-M. Bender, Alex Schuddeboom, Adrian J. McDonald, and Øyvind Seland

July 22, 2022

Dear anonymous referee,

Thank you for your insightful comments. Please find our response below. In the following text, the original comments are in **bold**, followed by our response. We do not provide a document marking differences between the original and revised manuscripts, because the changes were too substantial for it to make sense.

In the revised manuscript we use a new version of the ANN which performs pixel-wise classification. We hope that this change will address some of the main concerns. We also present a classification for 10 and 27 cloud types, which provides more nuanced results than what would be possible with more traditional cloud classifications. We changed the manuscript title to 'Machine learning of cloud types in satellite observations and climate models' to emphasise the new method of cloud classification over of the implications on climate sensitivity.

Kind regards,

Dr. Peter Kuma on behalf of the authors

**Summary**

**The authors train a supervised deep convolutional artificial neural network to predict the frequency of occurrence of four human-observed cloud types based on the CERES-measured top of atmosphere longwave and shortwave radiation fields over a large 4000 km x 4000 km region. After validating its ability to reproduce the observed cloud types on data withheld during training and comparing their results with an independent cloud categorization analysis, they apply the algorithm to climate model output, thereby allowing them to evaluate the fidelity with which models simulate the various cloud type occurrence frequencies. Model skill in simulating these current-climate cloud occurrences is assessed in light of the model climate sensitivities (ECS and TCR) and cloud feedbacks, and it is found that more sensitive models tend to have smaller mean-state cloud errors, although the cloud feedback shows little relationship with mean-state cloud errors. The authors argue that the most likely explanation for their results is that high ECS is plausible, in contrast to recent expert assessments (Sherwood et al. 2020; Masson-Delmotte et al. 2021) I find the paper's overall goal to be interesting and worthwhile, but I had substantial difficulty following it in several sections and I believe the authors need to be a little more circumspect with the interpretation of the results. I recommend major revisions, as detailed below.**

**Comments**

**1. I had a very hard time following what was done in setting up, training, and validating the ANN. This includes both understanding it at the conceptual level and in many of the details. I think the authors need to begin Section 2.2 with a "30,000 foot" view of what they are trying to do, namely, predict the frequency of occurrence of 4 WMO cloud types within a 4000 km x 4000 km box based on the (spatial pattern of?) TOA radiative fluxes observed within that box. I think some interpretation of what information the ANN is learning from is needed. Is it the spatial pattern / orientation of SW and LW radiation within the region, the regional-average values, or something else entirely that provides the needed information? Are the SW and LW information equally important or does one band provide most of the information? Why is a deep convolutional artificial neural network needed in the first place; what is it providing that simpler methods would fail to yield? Table 2 and Algorithm 1 are utterly incomprehensible to me, and probably to a majority of readers of this journal. These details should probably go in a supplementary materials or an**

appendix, and they should be replaced with something more like schematics that give a sense of the basic workflow, how the ANN is set-up, how the data is split among training and testing, etc. All of these details are hard to extract from the paper.

We added Section 2.1 and Fig. 1 which give a broad overview of the method. The learning is based on spatial information in the samples. Because both SW and LW radiation provide important information about clouds, it can be expected that they are both important for the accuracy of the ANN, even though we could successfully train the ANN on LW radiation only (with worse accuracy). In Section 3.3 in the revised manuscript, we compare the method with a partitioning based on ISCCP (Rossow and Schiffer, 1991). In addition to four cloud types, we provide results for 10 and 27 cloud genera/species, which demonstrate how the method can be useful over more traditional methods. We removed Fig. 2 and Algorithm 1 because they are probably too technical and the same information can be found in the supplied code.

**2. Is it wise to use random selection for splitting the training and testing subsets? I would assume that there could be some autocorrelation in the data that would cause such an approach to overstate the skill relative to a situation in which, say, the (chronologically) first 80% of the data is used as training and the last 20% of the data is used for testing, etc.**

We trained the revised ANN with a validation dataset consisting of years 2007, 2012 and 2017, which were excluded from the training dataset.

**3. What does it mean that the ANN explains 47% of the variance? Variance in what? And where does this number come from; is it in a figure? On line 407, it is stated that this number is "relative to an uninformative predictor" but elsewhere it is not stated relative to anything. How do I interpret this?**

We do not include this type of metric in the revised manuscript. Instead, the ANN is trained using a loss function defined using log-likelihood (Section 2.3) and for validation of the ANN we provide plots of the IDD dataset in Fig. 3, discussed in Section 3.2.

**4. Figures 4 and 5: "Stratiform" is misspelled**

We corrected this in the revised manuscript.

**5. Section 3.3.: How did you calculate these joint histograms? I can't find any details on how this is done or what data is used in the methods section.**

We added Section 2.4 explaining how these histograms were produced.

**6. Section 3.4: I am completely lost in this section on comparing to MODIS and ISCCP cloud clusters. How is the ANN now being generated on a 5x5 degree grid when previously the highest resolution is 4000 km x 4000 km? What exactly are you showing in Figure 7? I assume the reader has to be familiar with Schuddeboom et al. (2018) and McDonald and Parsons (2018) to understand this, but how many readers will be? I read this several times and I simply cannot wrap my head around what is being done here, other than a vague sanity check that what the ANN calls "high" , "stratiform", etc. is consistent with independent cloud clustering methods. I recommend a complete re-write of this section keeping in mind that the average reader is not familiar with these other studies.**

We rewrote this section to better explain the comparison and give enough context to understand the comparison without reading the referenced papers.

With the original ANN, even though the samples were 4000×4000 km, geographical distribution on 5×5° grid was produced by counting the contribution of every overlying sample to a grid cell. However, the effective resolution was much lower.

With the revised ANN, the classification is done on a 2.5×2.5° grid, but the effective resolution is about 5°.

The figure shows cloud clusters generated using SOM in the referenced papers, and how they co-occur with the ANN cloud types. In the revised manuscript we give a more detailed explanation in the caption and also specify cluster numbers in the plots.

**7. Section 3.5: I do not see any justification for regressing the observed cloud types on global mean surface temperature during the brief CERES record that (1) it is likely dominated by internal variability that is not directly relevant to the long-term cloud feedback and (2) likely includes effects related to changes in aerosols and other non-CO2 forcings. Placing these results side-by-side with the abrupt-4xCO2 cloud changes is misleading and not a robust evaluation of models. You already note that this is not a "reliable observational reference", so I wonder why you did it.**

We do not include CERES in the revised figure for abrupt-4xCO2. It was included in the original figure because within the error bars it was still possible to tell if a particular cloud type was increasing or decreasing during the instrumental record of CERES.

**A more minor point: the abrupt-4xCO2 simulation does not occur during a particular time in the historical record (noted here as 1850-1949) but rather to an arbitrary 150-year period whenever the modeling center decided to branch from its piControl simulation. So I think you meant to say simply that you used data from the 150-year experiment.**

We clarified this in the revised manuscript.

**8. Bayes factors: Maybe I am just ignorant, but this is the first time I had ever seen these numbers for significance testing. Perhaps other readers will also be clueless about what these numbers mean. Please provide some brief explanation when these first appear in the text, and discuss in more detail their meaning in the appendix.**

In the revised manuscript, we specify probability of the null hypothesis instead of Bayes factor, also calculated using Bayesian model comparison.

**9. Lines 321-323: The choice to report the ECS values for RMSE values below an arbitrary threshold (2.4%) is a little egregious, given that if the threshold for what is considered "low" RMSE is relaxed only slightly (to say 5%), the entire range of ECS values is now supported. Ditto for the "high" RMSE values: If you take all models with higher-than-average or even probably the models in the top 10 percentile of RMSE, you will include the high-ECS CanESM5 model.**

With the revised ANN, all models with ECS greater than 4 K have RMSE below 3% (a group 9 models of 18). All but three models with ECS lower than 4 K have RMSE above 3% (a group 9 models of 18). In the revised manuscript we fixed a technical issue with processing of CanESM5 and NorESM2 data. As a result, they are much more consistent with the rest of the models.

**10. Figure 10b: I wonder how large the Bayes factor would be if models from the same modeling center were averaged together before computing significance. Would the relationships derived in the paper between cloud occurrence RMSE and ECS remain so strong once the 3 UKMO models, 2 CNRM models, 2 INMCM models, 3 IPSL models, and 3 MPI models are combined? This would represent a substantial decrease in sample size from 18 to 10.**

We performed a sensitivity test with only 10 models as suggested. The null hypothesis model (no linear relationship) probability remains less likely than the alternative model for ECS and TCR, but not cloud feedback, with $P(M_0) = 0.29, 0.29, 0.62$ respectively. We included this information in Section 3.5 in the revised manuscript.

**11. Lines 337-340: I find this reasoning for why simulating good mean-state clouds should translate to simulating good clouds in a future warmed state to be dubious. It is likely that clouds will inhabit an environment with different conditions in the future (e.g., one with higher SSTs, stronger inversions, and a sharper moisture contrast between the boundary layer and free-troposphere) – refer to the cloud controlling factor literature (Bretherton 2015; Klein et al. 2017).**

We skipped this part of discussion in the revised manuscript.

**12. Line 347: "lower ECS" is a little misleading, as I think you mean lower than the highest ECS values in CMIP models, but not lower than, say, the canonical IPCC range.**

We now explicitly specify low ECS as < 4 K and high ECS as > 4 K (in the context of our model ensemble) everywhere in the text.

**13. Lines 353-354: In my opinion the simplest / most likely explanation is neither of these, but rather that you are looking at a very small sample size of models (especially once you combine closely related models from the same center) and spurious correlations can occur. Perhaps more importantly, I am led to doubt the robustness of the correlation with ECS because the correlation with cloud feedback is poor (Figure 10d). How are we to believe that accurately simulating mean-state clouds translates to a better representation of ECS if the most obvious intermediary (cloud feedback) shows no relationship with mean-state cloud quality?**

In our new results the statistical relationship was strengthened ($P(M_0) = 2 \times 10^{-3}$, $9 \times 10^{-3}$ and $1 \times 10^{-2}$) for ECS, TCR and cloud feedback due to improvements in the ANN. In the revised manuscript we include results from the sensitivity test with a smaller set of unrelated models and we mention this limitation in the discussion and conclusions.

Including all available models in an ensemble is a relatively common practice (e.g. with emergent constraints in Schlund et al., 2020, but also in various multi-model statistics in AR6) as well as using perturbations of a single model. CMIP5 and CMIP6 models are all generally highly code-dependent because of historical code sharing within and between modeling centres, and this impacts many model ensemble studies.

We reformulated the discussion, conclusions and abstract in a way that makes it more clear that this limitation exists.

**14. Lines 400-404: I don't understand what is being suggested here or how it could be used in concert with the techniques employed in this study.**

We changed this to: 'Potentially, other satellite products which provide TOA radiance information could be used instead of CERES, such as ISCCP, Multi-angle Imaging SpectroRadiometer (MISR), MODIS, CloudSat or the Cloud-Aerosol Lidar and Infrared Pathfinder Satellite Observation (CALIPSO). Because an equivalent physical quantity needs to be provided from a model for the ANN to be applicable on both, a satellite simulator such as the Cloud Feedback Model Intercomparison Project (CFMIP) Observation Simulator Package (COSP) could be used to calculate such an equivalent quantity from the model output.'

**15. Lines 432-435: My read of Zelinka et al (2022) is that the quality of present-day cloud representation has very little bearing on the quality of its cloud feedback (see their Figure 4b). Seems worth mentioning this, rather than the weaker statement that it is an open question.**

In their Fig. 4a, they show that it is related to cloud feedback. Their Fig. 4b shows little relation to cloud feedback RMSE (with respect to cloud feedback as assessed by Sherwood et al., 2020). Therefore, they show a similar result as us in their Fig. 4a, compared with our Fig. 9d in the revised manuscript. In the revised manuscript, we added: 'They call this an open question for future research, although they note that "model parameters driving variance in mean-state extratropical cloud-radiative effect across members of the HadGEM3-GA7.05 perturbed physics ensemble differ from those driving the variance in its cloud feedback (Tsushima et al., 2020)." '

**16. Lines 435-440: Somewhere around here it may bear mentioning the notion that emergent constraints based on mean-state climatological observables (like the occurrence frequency of the 4 cloud types in this study) are generally less useful or robust than those that narrow in on processes relevant to the climate change phenomenon of interest (Klein and Hall 2015; Hall et al. 2019)**

In the discussion and conclusions we add: 'We suggest that our results about high sensitivity models being more correct in their cloud type representation should be considered with caution due to the novelty of the method, the size and cross-correlations in the model ensemble and need for a physical explanation.'

With the revised ANN we show that in the abrupt-4xCO2 experiment, low ECS (< 4 K) models tend to simulate increasing stratiform clouds and decreasing cumuliform clouds, while high ECS (> 4 K) models tend to simulate the opposite (Fig. 6b). Due to the lower optical depth of cumuliform clouds, this may be a physical reason partly responsible for the difference between the low and high ECS models.

---

## Referee Report (RR1)

**Review of "Machine learning of cloud types in satellite observations and climate models"**
**by Kuma et al**
**MS No.: acp-2022-184**

**Summary**
The authors have revised the manuscript to address the reviewers' previous comments, which has generally improved the paper. This includes dialing back the conclusions regarding implications for climate sensitivity, adding much more detail to help readers understand the methodology, and performing the analysis at a pixel-by-pixel level. However, there remain several things that need to be addressed before the paper is suitable for publication.

**Major Comments**

- I am still quite unclear on how the ANN works. The input TOA flux data are used to predict the probability of the 4 cloud types at each pixel in each 16x16 pixel domain, but the ground truth labels in most cases will only occupy a small portion of the domain where the IDD stations are. How can the pixels with no ground truth learn anything from the CERES TOA fluxes?

- I don't understand what is meant by the "application phase." Does this refer to the phase when you deploy the trained ANN to make predictions on unseen data? If so, then why does it only use 20 random samples per day rather than all of the TOA radiation data?

- Why are there no figures demonstrating the skill of the ANN in predicting unseen data? I see that the ANN is trained on CERES and IDD data in years 2004, 2005, 2007 and 2009–2017, with years 2007, 2012 and 2017 used as a validation dataset. My understanding of how ML studies are typically done is that the data is split into three categories: training, validation, and testing. It appears as though here you have used all of the data for training and validation, but did not reserve some data for doing out-of-sample testing. How can we be sure that the ANN works well on unseen data and has not over-fit to the training data?

- All of the analysis is basically in frequency of occurrence space rather than in within-regime cloud property space. But surely the latter should be a large part of the story. A model could for example get the frequency of occurrence of each regime perfectly right but the cloud properties (cloud fraction, albedo, altitude) within the 4 regimes could be biased. Is there a reason that within-regime cloud properties are not evaluated as well as frequencies of occurrence?

**Minor Comments**

L8: delete "a" before "top"

L55: there are issues with subject-verb agreement ("they…is…has…")

L99: I don't understand this statement about grouping together multiple cloud genera, since throughout the paper the results for 10- and 27-type classifications are also shown.

Section 2: I dislike the organization here. It goes from Methods description (Section 2.1) to Data used (Section 2.2) back to Methods description (Section 2.3). Why not put the Data section first?

Figure 2: Suggest calling the radiation fields what they actually are (normalized CRE) rather than "reflected TOA radiation" (colorbar) or "shortwave and longwave radiation (caption). Do the 4 cloud type maps have to sum to 100%? It doesn't look like this is the case. Is there a clear-sky probability?

L291: "histograms" should be singular

L333-334: Why are you reporting the years like this? It doesn't make any sense, as I noted in my previous review.

Figures 18-19: These are completely ineffective and uninformative figures that should be removed. Are responses of the individual 27 cloud types really trustworthy? Even if they are, is examining responses with this level of granularity bringing any new any insights? I doubt it.

L345: "the the"

L358: What is the P value for high clouds? Which value of P marks the transition from "statistically identifiable" to "not statistically identifiable"? In the previous paper, the Bayes factor (ratio of the two probabilities) was reported, but now just the probability of the null hypothesis is reported. Is there a reason for this?

L397-414: I don't see the value in this discussion. All of the options discussed would still not allow for an unambiguous estimate of the response of clouds to global warming relevant for ECS. This is because of spurious trends in the datasets, the influence of factors other than just global warming (aerosols, most notably) on the trends, the fact that the observed warming pattern over the satellite period is very different from that expected in response to CO2, and other things. I am not aware of any TOA flux measurements on MISR, MODIS, CloudSat, or Calipso, so I'm not sure why those would be used instead of a radiometer like CERES. The idea of running a COSP simulator in a model to generate fields to be run through an ANN to tell you about cloud types seems really bizarre since COSP is already providing detailed information about cloud types. I suggest deleting this paragraph.

L428: I'm confused. Zelinka et al call what an open question? The previous sentence just looks like a statement of fact – that better present-day cloud properties is associated with larger cloud feedback, similar to what is found in this study. What is the question? In the next line, is it really necessary to directly quote that paper (Zelinka et al) for a fairly mundane statement summarizing the results from another paper (Tsushima et al)? Usually quotes would be reserved for something where the exact phrasing is vital or compelling.

L438: I believe the correct citation is "Jiménez-de-la-Cuesta" and Mauritsen (2019)

L343-445: This paragraph seems to be all over the place and it is not clear what point you are trying to make.

Appendix A: suggest telling the reader what values of P or Bayes factor represent statistical significance (e.g., something analogous to p values being less than 0.05 for a statistically significant result at 95% confidence)

Figure B1: I still don't really know what I am looking at here. Is there a way of showing the reader what "perfect" validation looks like? I have no idea what "right" or "wrong" looks like.

---

## Author Response (AR2)

**Response to Referee 1 on revised manuscript 'Machine learning of cloud types in satellite observations and climate models'**

Peter Kuma, Frida A.-M. Bender, Alex Schuddeboom, Adrian J. McDonald, and Øyvind Seland

November 2, 2022

Dear Dr. Steven Sherwood,

Thank you very much for the second round of comments. Please find our response below. In the following text, the original comments are in **bold**, followed by our response.

Kind regards,

Dr. Peter Kuma on behalf of the authors

**1. As raised by both reviewers of the original paper, it was very odd that the performance on the test data was slightly higher than on the training data. Now that the authors have more clearly described their approach, it is clear that they did not use independent test and training data. The test data come from a subset of the same years as used in training. The authors choose instances at random from within those years, but due to spatio-temporal autocorrelation of the data (and the fact that nothing is said about any effort to ensure that the training and test data tiles don't overlap), this will not result in an independent test. The authors should either redo their analysis with non-overlapping training and test periods, or else they should just use all the available data for training and explain that they did not do an independent test at all (this would be unsatisfactory and against usual practice, but would at least be an accurate description).**

The sentence in our revised manuscript describing the validation was wrong (L260, 'We trained the ANN on CERES and IDD data in years 2004, 2005, 2007 and 2009–2017, with years 2007, 2012 and 2017 used as a validation dataset, representing 25% of the total number of years.'). The training and validation years were strictly separate, with the training dataset consisting of years 2004, 2005, 2009–2011, 2013–2016, and 2018–2020, and the validation dataset consisting of years 2007, 2012 and 2017. We changed the sentence to: 'We trained the ANN on CERES and IDD data in years 2004, 2005, 2009–2011, 2013–2016 and 2018–2020, with years 2007, 2012 and 2017 used as a validation dataset, representing 20% of the total number of years.' We apologise for this error.

The validation dataset was used in the training only as a stopping criterion – the training process was stopped when the loss function on the validation dataset was not improved for three consecutive training rounds. Therefore, the input of the validation dataset on the training process was likely negligible.

In the second revised manuscript, we also perform validation by training the ANN on the training data excluding station data from a number of geographical locations – quarters of the globe (north-east, north-west, south-east, south-west), as well as four smaller regions over North Atlantic, East Asia, Oceania and South America. In this way, we test the ability of the ANN to predict cloud occurrence over stations for which no information was supplied during the training, we well as temporally to years in included in the training (2007, 2012 and 2017). The new validation is described in Sect. 3.3 and 4.1. The new validation includes comparison of between the ANN and the IDD stations in the regions excluded from the training in terms of RMSE of daily means and all-time means, and a receiver operating characteristic (ROC).

**2. Related to (1): the authors do not present any quantification of the skill of their ANN on the IDD data. The only skill scores reported compare the ANN classifications based on GCM or reanalysis radiances to those based on CERES observed radiances. This is legitimate as a way of comparing models to obs, but does not assure us that the basis of comparison is physically meaningful or that the station data have actually added anything. We can roughly assess the skill by comparing the first two rows of Fig. 3, but this is hard to see, and only addresses the skill of time-averaged fields whereas the method should also capture time variations (does it?). They do not assess skill on the micro-classifications**

**at all, yet present some results on those later. If they are going to present such results they need to show whether the ANN can actually distinguish any of those cloud classes.**

As mentioned in the point above, we added a new validation which compares the ANN with IDD station data not included in the training. We also determine the accuracy on daily scales as RMSE and ROC. We do not relay on accuracy of the ANN on daily scales for the main results of the study. All presented figures are long-term means. To some extent Fig. 5 (cloud optical depth–cloud top pressure; in the second revised manuscript Fig. 8) and Fig. B1 (comparison with SOM) depend on daily scale accuracy, even though they present the information as long-term means. The other main figures (Fig. 3, 4, 6–8 and 9; in the second revised manuscript Fig. 6, 7, 9–12) are based entirely on long-term means, and no daily scale information is compared between the ANN applied on models and the ANN applied on CERES.

**3. The authors note that one prior study (Shuddeboum and MacDonald 2021) that was similar did not find any relationship to ECS. In Appendix B they seem to explore in detail the alternate classification system, but then do not say anything about why this discrepancy occurred. Can they work out based on this analysis why the answers were different?**

The strongest factor might be the small set of models analysed in Schuddeboom and McDonald (2021) – only 8 models. The two cloud classifications are not the same by definition, and therefore there is no reason to think that they have to show the same relationship to ECS. We performed the comparison between the two classifications (1) as a sanity check for our classification scheme, and (2) to provide context for the new classification, alongside a comparison of the cloud top pressure – cloud optical depth histograms with the ISCCP classification (Sect. 4.3). The classification schemes represent particular decomposition of the radiation fields from which they are derived. Different statistical associations can be identified depending on the type of decomposition, but of course such associations might not always imply a causal physical relationship. In their Fig. 8 they analyse association between global compensating error in terms of SW CRE of the cloud regimes and ECS. This is conceptually different from our analysis of association of the RMSE of cloud type occurrence probability and ECS.

We added a sentence in the discussion and conclusions: 'The reason why their result is different from ours might be due to a number of factors, such as a small number of models analysed by Schuddeboom and McDonald (2021), a different set of models, their focus on SW CRE errors vs. our focus on the RMSE of cloud type occurrence probability, and a very different cloud classification method.'

**4. The fact that the authors chose to assign a 50/50 prior to a null model vs a linear effect model is very important and should be stated in the main text, not just Appendix A. Arguably since the linear-effect model is more complex and carries an additional parameter, it should be assigned a much lower prior, which would change the results (especially using only one model per modelling centre, in which case the most probable hypothesis would switch from the linear-effect model to the null model). Just by eye, the relationships in Fig. 9b-d do not look strong to me so I think the probabilities being estimated by the authors are being skewed by the statistical assumptions. I do agree that the stratiform correlation in Fig. 9a looks impressive (though hard to see due to the colour) but even in this case I would surely not give it a 99.99% chance of being a real relationship, as the authors do!! 80-90% maybe. Due to the sensitivity of their probabilities to the arbitrary choices listed in Appendix A, I think it would benefit readers to also quote a standard correlation coefficient for the panels in Fig. 9 at least.**

We added a sentence 'We note that for the statistical analysis we used priors for the null and alternative models equal to 0.5 (Appendix A).' to the first paragraph where we mention the statistical analysis (Sect. 4.6). Using equal priors is a common practice. In this case, it does not favour the alternative model because it has more free parameters (three parameters: slope, intersect and error spread) than the null model (two parameters: mean and error spread), which implicitly penalises the alternative model. To supplement the Bayesian statistical analysis, in the second revised manuscript in Fig. 11 and 12 we also include the coefficient of determination, correlation coefficient, 95% confidence interval of the slope, Bayesian probability that the slope is greater or lower than zero, and the p-value of a z-test for the difference of means of two groups of models in the bottom 50% and top 50% of ECS, TCR and cloud feedback. The result of the z-test is $3 \times 10^{-6}$, $3 \times 10^{-6}$ and $4 \times 10^{-4}$ for ECS, TCR and the cloud feedback, respectively.

A z-test for the stratiform cloud type in Fig. 9a in the first revised manuscript (Fig. 11d in the second revised manuscript) for the difference of mean between groups of models with ECS below 4 K and above is $3 \times 10^{-5}$.

What is not included in the statistical estimates is the impact of model interdependence, which is hard to quantify objectively and would have the largest impact on reducing confidence in the identified relationships. As mentioned earlier, this is not an uncommon practice and also applies to other similar studies such as Schlund et al. (2020). We repeated the statistical tests with

a limited set of independent models (AWI-ESM-1-1-LR, CNRM-CM6-1-HR, CanESM5, EC-Earth3P, INM-CM5-0, IPSL-CM6A-LR, MPI-ESM1-2-HR, MRI-ESM2-0, NorESM2-LM and UKESM1-0-LL). The p-value of the z-test is $7 \times 10^{-3}$, $7 \times 10^{-3}$ and $6 \times 10^{-2}$ for ECS, TCR and the cloud feedback (Fig. S9) for the 27 cloud types, and $5 \times 10^{-3}$, $5 \times 10^{-3}$ and $5 \times 10^{-2}$ for the four cloud types.

**5. Given that Figs. 6-8 don't really show interesting trends with ECS or indeed anything that really stands out, I'd strongly consider eliminating or simplifying them to show whatever it is the authors want to show. Fig. 8 is especially hard if not impossible to see anything in, especially given my comments under (2) above.**

In Fig. 6a (Fig. 9a in the second revised manuscript) the error in the cumuliform cloud type stands out quite starkly, together with its dependence on ECS. We performed a z-test for the difference between the cumuliform cloud type error between a group of models with ECS < 4 K and a group of models with ECS >= 4, and the p-value is $3 \times 10^{-4}$. We included results of the z-test in the second revised manuscript (Fig. 9a). Fig. 6b (Fig. 9b in the second revised manuscript) for example shows that the stratiform cloud type tends to be increasing with GMST in low ECS models and constant/decreasing in models with high ECS. This relationship is tested statistically in Fig. 9a (Fig. 11 in the second revised manuscript). We think this justifies keeping the figure.

We moved Fig. 7 and 8 to the supplement and made visual improvements in the figures to improve readability. Some useful information that is shown in these figures is that the error in the cumuliform cloud type is attributed to the Cu and Cu+Sc type (and not, for example, Cb).

**Minor issues:**

**The text at line 121 seems to say that only two radiation values were used for each training sample, when in fact it is two vectors of 256 normalised radiation fluxes.**

We changed the sentence to 'In detail, the ANN input were samples consisting of two channels of SW and LW radiation (256 values for each channel in 16 by 16 pixel samples), ...'

**The text at line 434-445 seems to repeat things that should have been covered in the introduction. All text in this section should do is refer back and say how the presented findings relate to past work. Actually I'm not sure most of the work described here is even relevant to the paper.**

We removed the first sentence of the paragraph and integrated the rest into the first two paragraphs of the introduction.

**Response to Referee 2 on revised manuscript 'Machine learning of cloud types in satellite observations and climate models'**

Peter Kuma, Frida A.-M. Bender, Alex Schuddeboom, Adrian J. McDonald, and Øyvind Seland

November 2, 2022

Dear anonymous referee,

Thank you very much for the second round of comments. Please find our response below. In the following text, the original comments are in **bold**, followed by our response.

Kind regards,

Dr. Peter Kuma on behalf of the authors

**Summary**

**The authors have revised the manuscript to address the reviewers' previous comments, which has generally improved the paper. This includes dialing back the conclusions regarding implications for climate sensitivity, adding much more detail to help readers understand the methodology, and performing the analysis at a pixel-by-pixel level. However, there remain several things that need to be addressed before the paper is suitable for publication.**

**Major Comments**

**\* I am still quite unclear on how the ANN works. The input TOA flux data are used to predict the probability of the 4 cloud types at each pixel in each 16x16 pixel domain, but the ground truth labels in most cases will only occupy a small portion of the domain where the IDD stations are. How can the pixels with no ground truth learn anything from the CERES TOA fluxes?**

The reference IDD station data are only available on some pixels of a sample. The ANN uses a U-Net type of network to quantify the cloud type occurrence probability at every pixel of the sample. The loss function is only calculated from those pixels where an IDD station is available. The other pixels do not have any reference during the training process. However, the samples are large enough to contain synoptic-scale cloud patterns in the normalised CRE fields. Therefore, the ANN can learn to recognise these patterns and associate them with certain cloud types. In addition, the value of the normalised SW and LW CRE at any given pixel also provides information about the possible cloud type. Because the samples are centred at random geographical points during the training process, an IDD station will end up located in different random pixels of the sample (on the $16 \times 16$ grid). This provides training for all pixels of the sample on the $16 \times 16$ grid. The U-Net design of ANNs trains coefficients at a number of size scales of the sample in a number of downsampling and upsampling steps. The coefficients are therefore affecting the whole sample and not only one pixel. In this way, the ANN is able to quantify all pixels of the sample, despite only having training data for a limited number of pixels. Overfitting is prevented by the use of a validation dataset and a stopping criterion, which stops the training process once the independent validation dataset stops improving for three consecutive training steps.

In the second revised manuscript, in Fig. 3 we show that the ANN is able to extrapolate synoptic-scale and global-scale patterns to regions for which it had no training information. In Sect. 4.1 we provide more validation which also includes evaluation over four regions excluded from training: North Atlantic, East Asia, Oceania and South America.

**\* I don't understand what is meant by the "application phase." Does this refer to the phase when you deploy the trained ANN to make predictions on unseen data? If so, then why does it only use 20 random samples per day rather than all of the TOA radiation data?**

Yes, the application phase refers to making predictions from unseen CERES or model normalised CRE data. The input in the application has to have the same format as during the training phase, i.e. must be a $16 \times 16$ grid of SW and LW normalised CRE. This means we cannot supply the whole normalised CRE fields to ANN covering the entire globe at once. The reasons why we use samples instead of for example the fields on a regular longitude–latitude grid is to avoid issues due to spatial transformation in a geographical projection. Every geographical projection causes deformation which can cause biases in the prediction. We limit this by centring the local geographical projection in the centre of the samples. We also want to prevent the ANN training from recognising geographical patterns such as coastlines instead of training only on the cloud fields. This cannot be fully prevented, but it can at least partially mitigated by training it on randomly located samples.

The reason why we only use 20 samples and not more is due to performance reasons. 20 samples already cover a substantial part of the globe on a single day.

**\* Why are there no figures demonstrating the skill of the ANN in predicting unseen data? I see that the ANN is trained on CERES and IDD data in years 2004, 2005, 2007 and 2009–2017, with years 2007, 2012 and 2017 used as a validation dataset. My understanding of how ML studies are typically done is that the data is split into three categories: training, validation, and testing. It appears as though here you have used all of the data for training and validation, but did not reserve some data for doing out-of-sample testing. How can we be sure that the ANN works well on unseen data and has not over-fit to the training data?**

The sentence in our revised manuscript describing the validation was wrong (L260, 'We trained the ANN on CERES and IDD data in years 2004, 2005, 2007 and 2009–2017, with years 2007, 2012 and 2017 used as a validation dataset, representing 25% of the total number of years.'). The training and validation years were strictly separate, with the training dataset consisting of years 2004, 2005, 2009–2011, 2013–2016, and 2018–2020, and the validation dataset consisting of years 2007, 2012 and 2017. We changed the sentence to: 'We trained the ANN on CERES and IDD data in years 2004, 2005, 2009–2011, 2013–2016 and 2018–2020, with years 2007, 2012 and 2017 used as a validation dataset, representing 20% of the total number of years.' We apologise for this error.

The validation dataset was used in the training only as a stopping criterion to prevent overfitting – the training process was stopped when the loss function on the validation dataset was not improved for three consecutive training rounds. Therefore, the input of the validation dataset on the training process was likely negligible.

As mentioned in the point above, we included more extensive validation in Sect. 4.1 which covers temporal and geographical extrapolation.

**\* All of the analysis is basically in frequency of occurrence space rather than in within-regime cloud property space. But surely the latter should be a large part of the story. A model could for example get the frequency of occurrence of each regime perfectly right but the cloud properties (cloud fraction, albedo, altitude) within the 4 regimes could be biased. Is there a reason that within-regime cloud properties are not evaluated as well as frequencies of occurrence?**

To some extent the cloud type occurrence probability estimated by the ANN already contains information about cloud properties such as cloud fraction, cloud optical depth and altitude. Therefore, an evaluation of the cloud type occurrence probability is also an evaluation of the said properties, even though they are not treated separately.

We added Sect. 4.5 which analyses cloud properties (cloud fraction, cloud top pressure and cloud optical depth) in models relative to CERES as a function of cloud type. The reason why we did not analyse these were both time constraints and the fact that these properties are relatively hard to compare across models and observations. Some of the difficulty arises from potentially different definitions of cloud fraction in models, and if an instrument simulator is used, it should be consistent for all of the models/observations. This is hard to achieve when it is not possible to run the models and instead we have to rely on what is available in the archives. Even if some of these properties may be consistent in CMIP models, MERRA-2 and ERA5 use other conventions. Normalised CRE which was used in the rest of our analysis was chosen as an input because it is consistently defined in models and observations and known with relatively high accuracy from satellite observations.

**Minor Comments**

**L8: delete "a" before "top"**

We corrected this in the revised manuscript.

**L55: there are issues with subject-verb agreement ("they...is...has...")**

We reformulated this sentence to: 'Therefore, they can be used as a metric for model evaluation which, unlike metrics based on more synthetically-derived cloud classes, is easy to understand and has a very long observational record.' The singular here refers to the metric.

**L99: I don't understand this statement about grouping together multiple cloud genera, since throughout the paper the results for 10- and 27-type classifications are also shown.**

This was not meant to be a statement referring to all of the classifications, only the classifications of 4 and 10 cloud types. We clarified this in the revised manuscript: "For practical purposes, in our analysis we grouped together multiple cloud genera to a smaller number of 'cloud types' (four and ten), in addition to using the full set of 27 WMO cloud genera."

**Section 2: I dislike the organization here. It goes from Methods description (Section 2.1) to Data used (Section 2.2) back to Methods description (Section 2.3). Why not put the Data section first?**

We re-ordered the sections so that Data (Sect. 2 in the second revised manuscript) comes before Methods (Sect. 3 in the second revised manuscript). The reason for the original order was that the outline section might be useful to the reader before reading the data section for understanding the general aim. The outline section was added in response to a previous comment asking for a general overview of the method before describing the details.

**Figure 2: Suggest calling the radiation fields what they actually are (normalized CRE) rather than "reflected TOA radiation" (colourbar) or "shortwave and longwave radiation (caption). Do the 4 cloud type maps have to sum to 100%? It doesn't look like this is the case. Is there a clear-sky probability?**

We changed the colourbar label to 'Normalised CRE (%)' and the caption to 'Shown is normalised shortwave (SW) and longwave (LW) cloud radiative effect (CRE), and the probability of cloud type occurrence calculated by the ANN for the classification into four cloud types.'

The cloud type occurrence probability does not generally sum up to 100% because the cloud types are not mutually exclusive in the station measurements. Each WMO station report can identify up to three cloud genus/species in the three SYNOP/BUOY fields $C_L$, $C_M$ and $C_H$. Clear-sky probability was not predicted by the ANN. In the Fig. 2 (Fig. 1 in the second revised manuscript) caption, we added: 'Note that the cloud types are not mutually exclusive, and therefore do not have to sum to 100%.'

**L291: "histograms" should be singular**

The plural refers to the four histogram in Fig. 5b–e (Fig. 8b–e in the second revised manuscript).

**L333-334: Why are you reporting the years like this? It doesn't make any sense, as I noted in my previous review.**

The sentence in question: 'It was calculated from the first 100 years for CMIP abrupt-4xCO2, with the exception of two models for which the first 100 years were not available: for MPI-ESM-LR years 1850–1869 and 1970–1989 were used, and for MRI-CGCM3 years 1851–1870 and 1971–1990 were used (1850 is the start of the abrupt-4xCO2 experiment, and the time period is not supposed to correspond to reality).'

The years here correspond to the years as in the product files. This is to allow anyone to reproduce the results. We understand that these are not real years, but rather arbitrarily chosen years which depend on the chosen start of the abrupt-4xCO2 experiment in the model run. Some models do not provide all years in the time period between the first and 100[th] year of the abrupt-4xCO2 experiment, but they still provide limited time periods in the beginning and end of the experiment. For these models, we explicitly listed those time periods (1850–1869 and 1970–1989 in MPI-ESM-LR, and 1851–1870 and 1971–1990 in MRI-CGCM3). Again those are not real years, but rather what is in the *time* variable in the product files.

We reformulated the sentence to make it more clear: 'It was calculated from year 1 to 100 of the CMIP abrupt-4xCO2 experiment. Some models do not provide all years in this time period. These models are MPI-ESM-LR, for which we used years 1850–1869 and 1970–1989, as in the *time* variable of the product files, and MRI-CGCM3, for which we used years 1851–1870 and 1971–1990. These years do not correspond to real years, but rather an arbitrary time period starting with 1850 used for the abrupt-4xCO2 experiment in these models.'

**Figures 18-19: These are completely ineffective and uninformative figures that should be removed. Are responses of the individual 27 cloud types really trustworthy? Even if they are, is examining responses with this level of granularity bringing any new any insights? I doubt it.**

The figures were added in response to a comment that an analysis of only four cloud types is not sufficiently different from other cloud classifications. Some readers might still be interested in the large sets of cloud types. The supplementary plots 'geo_cto_historical_10_x.png' and 'geo_cto_historical_27_x.png' show that the larger sets of cloud types have geographically distinct distribution, and therefore it can be expected that different physical processes are involved. They also have clearly different cloud optical depth – cloud top pressure distribution. For atmospheric modelling, it is an important question whether certain physical processes leading to the formation of a particular cloud type is well represented. One of the main points of comparison between models and observations is to provide information for modellers, which they could use for model improvement. More granular results can help to identify particular processes and geographical locations which are responsible for biases. A new insight from the classification for 10 cloud types is that the cumuliform cloud type bias is dominated by the Cu cloud genus (and not, for example Cb, which is also in the same group in the classification of four cloud types). We recognise that Fig. 9 and 10 may be too detailed for most readers. In the revised manuscript, we moved them to the supplementary information. We also made the plots colour blindness friendly in line with the journal guidelines.

**L345: "the the"**

We corrected this in the revised manuscript (as well as other instances of the same mistake).

**L358: What is the P value for high clouds? Which value of P marks the transition from "statistically identifiable" to "not statistically identifiable"? In the previous paper, the Bayes factor (ratio of the two probabilities) was reported, but now just the probability of the null hypothesis is reported. Is there a reason for this?**

We just want to clarify first that 'P(...)' in the manuscript refers to posterior probability and not a p-value. We used a value of 5% as a cutoff for statistically identifiable, which is commonly used with p-values and confidence intervals, i.e. 95% probability that the null hypothesis can be rejected. Because of the deficiencies of p-values (Colquhoun, 2014), p-values are usually weaker than the actual probability of the null hypothesis at the same threshold.

$P(M_0)$ for the high cloud type was 0.7. This value is included in Fig. 11 in the second revised manuscript.

Because we use prior probability of 0.5 for the null and alternative hypotheses, the Bayes factor (BF) is simply related to the probability of $M_0$ as $P(M_0) = BF/(1 + BF)$. We did not include BF in the first revised manuscript in response to referee comments noting that not many readers are going to be familiar with BF. In the second revised manuscript, we include multiple other measures of statistical association and significance in Fig. 11 and Fig. 12 in the second revised manuscript (the coefficient of determination, correlation coefficient, confidence interval of the slope, probability that the slope is greater/smaller than 0, and the p-value of a z-test for the difference of group means).

We clarified the sentence: 'Relation with the high cloud type was not statistically identifiable (probability below 5%).'. In Appendix A, we added: 'For statistical significance we assumed $P(M_0)$ below 0.05.'

**L397-414: I don't see the value in this discussion. All of the options discussed would still not allow for an unambiguous estimate of the response of clouds to global warming relevant for ECS. This is because of spurious trends in the datasets, the influence of factors other than just global warming (aerosols, most notably) on the trends, the fact that the observed warming pattern over the satellite period is very different from that expected in response to CO2, and other things. I am not aware of any TOA flux measurements on MISR, MODIS, CloudSat, or Calipso, so I'm not sure why those would be used instead of a radiometer like CERES. The idea of running a COSP simulator in a model to generate fields to be run through an ANN to tell you about cloud types seems really bizarre since COSP is already providing detailed information about cloud types. I suggest deleting this paragraph.**

The point of the discussion is not necessarily related to ECS. It would be useful to know the trend in the cloud types in the historical record regardless of ECS or aerosols, and if models can reproduce this trend in the *historical* CMIP experiment, which is based on real $CO_2$ concentration. Aerosol effects do not preclude such a comparison because they are commonly included in the *historical* experiment. Passive satellite instruments generally provide radiance measurements which can be potentially utilised with the ANN instead of normalised CRE – the ANN does not have to operate on flux which is corrected for angular and diurnal variation, but any kind of radiation field which carries information about clouds. We used normalised CRE because it is a very clean field to work with in terms of being corrected for angular and diurnal variations and accurately comparable between models and satellite observations. In the case of active instruments (CloudSat and CALIPSO), 2-dimensional fields (in time and height) of backscattered radiation could be used in the training process. For comparison with models, an equivalent output from the model would be needed, and such output can be provided by an instrument simulator. In the case of normalised CRE as used in our analysis, a simulator is not needed because models provide equivalent fields without the need for a simulator. The reason why radiation information other than TOA flux from CERES would be

used is because other satellites provide a more long-term records, and because active satellite like CloudSat and CALIPSO provide a qualitatively very different view of clouds – passive instruments can only see the top of clouds, whereas the active instruments can see the vertical structure of clouds. Our cloud types are different from other cloud classifications because they correspond directly to the WMO cloud genera, and COSP provides a different classification. Even though spurious trends in long-term satellite datasets such as the NOAA satellite series might preclude the evaluation of trends now, it is quite possible that in the future these datasets will be improved enough to determine trends (options for future research are what we are trying to lay out in this section). There is no reason why our method cannot be improved in the future by applying it on higher resolution radiation fields either from a satellite or future high-resolution climate models. Higher resolution could carry enough information to identify cloud genera much more precisely, and this is something which might not be possible with more traditional cloud classification methods.

We reformulated the paragraph to clarify these points and split it into two paragraphs.

**L428: I'm confused. Zelinka et al call what an open question? The previous sentence just looks like a statement of fact – that better present-day cloud properties is associated with larger cloud feedback, similar to what is found in this study. What is the question?**

From Zelinka et al. (2022), Sect. 3.4: 'While caution is necessary given the relatively small sample size, an important question is why better simulating present-day cloud properties is associated with larger cloud feedbacks. We leave this as an open question for future research.'

We reformulated the sentence to: 'They concluded that the explanation for this association is an open question for future research.'

**In the next line, is it really necessary to directly quote that paper (Zelinka et al) for a fairly mundane statement summarizing the results from another paper (Tsushima et al)? Usually quotes would be reserved for something where the exact phrasing is vital or compelling.**

We removed this part.

**L438: I believe the correct citation is "Jiménez-de-la-Cuesta" and Mauritsen (2019)**

We corrected this in the revised manuscript.

**L343-445: This paragraph seems to be all over the place and it is not clear what point you are trying to make.**

We removed this paragraph in the second revised manuscript because it referred to Fig. 7 and 8, which are in the supplementary information of the second revised manuscript.

**Appendix A: suggest telling the reader what values of P or Bayes factor represent statistical significance (e.g., something analogous to p values being less than 0.05 for a statistically significant result at 95% confidence)**

We added: 'For statistical significance we assumed $P(M_0)$ below 0.05.'

**Figure B1: I still don't really know what I am looking at here. Is there a way of showing the reader what "perfect" validation looks like? I have no idea what "right" or "wrong" looks like.**

The two classifications are different and a perfect match cannot be expected. Something close to the best result would be if the cloud types in CERES/ANN corresponded strongly to specific cloud regimes in ISCCP/SOM. A wrong result would be if the the CERES/ANN cloud types corresponded to many different ISCCP/SOM cloud regimes with about equal probability, or if they corresponded to regimes which are not physically consistent (in terms of cloud optical depth and cloud top pressure) with the cloud type. The comparison is included for providing a link to another cloud classification method rather than to strictly validate our results, because any two cloud classifications are going to be different depending on their definition.

**References**

Colquhoun David. 2014. An investigation of the false discovery rate and the misinterpretation of p-values. *R. Soc. open sci.* **1:** 140216140216. http://doi.org/10.1098/rsos.140216.

---

## Author Response (AR3)

**Response to Referee 2 on revised manuscript 'Machine learning of cloud types in satellite observations and climate models'**

Peter Kuma, Frida A.-M. Bender, Alex Schuddeboom, Adrian J. McDonald, and Øyvind Seland

December 5, 2022

Dear anonymous referee,

Thank you very much for the third round of comments. Please find our response below. In the following text, the original comments are in **bold**, followed by our response.

Kind regards,

Dr. Peter Kuma on behalf of the authors

I appreciate that the authors have considered my previous comments and am satisfied with their revisions. I recommend acceptance after considering my remaining minor comments:

• I doubt that many readers will be familiar with receiver operating characteristic plots. What does "specificity" mean? Please provide more information to help readers interpret the plot.

In the methods subsection on validation (Sect. 3.3), we added: 'For a validation of the ANN, we calculate the receiver operating characteristic (ROC) diagram (Sect. 4.1). An explanation of the diagram is given for example by Wilks (2019) in Chapter 9.4.6. The diagram shows sensitivity (the true positive rate) and specificity (the true negative rate) of the prediction for a set of choices of thresholds for a positive prediction, represented on the diagram by points on a curve. The area under curve (AUC) is calculated by integrating the area under a curve, and can be intepreted as a goodness of the prediction.'

In the caption of Fig. 5, we added: 'Shown is also the ROC of a random predictor. Sensitivity is the true positive rate (probability of a positive prediction if positive in reality), also called the hit rate. Specificity is the true negative rate (probability of a negative prediction if negative in reality). '1 - specificity' is also called the false alarm rate. Area under curve (AUC) of the ROCs is in the label.'

• L399: This could be a good place to remind the reader that the ANN is trained on ground-based cloud observations and so disagreement with satellite-based cloud data being larger for higher clouds might be expected.

We added: 'This may be due to the fact that our method is based on ground-based cloud observations, which are often not capable of identifying high and mid-level clouds. It can be expected that discrimination of high and mid-level clouds by the ANN is not as good as that of low clouds.'

• L442: "close to CERES"

We corrected this mistake.